# Implementation of RCIP scheme and its performance for 1D age computations in ice-sheet models

Fuyuki SAITO[1], Takashi OBASE[2], and Ayako ABE-OUCHI[2,1]

[1]Japan Agency for Marine-Earth Science and Technology (JAMSTEC), Yokohama, Japan
[2]Atmosphere Ocean Research Institute, University of Tokyo, Kashiwa, Japan

**Correspondence:** SAITO Fuyuki (saitofuyuki@jamstec.go.jp)

**Abstract.** Ice sheet age computations are formulated using an Eulerian advection equation, and there are many schemes that can be used to solve them numerically. Typically, these differ in numerical characteristics such as stability, accuracy, and diffusivity. Furthermore, although various methods have been presented for ice sheet age computations, the constrained interpolation profile method and its variants have not been examined in this context. The present study introduces one of its variants, a rational function-based constrained interpolation profile scheme (RCIP) to one-dimensional ice age computation and demonstrates its performance levels via comparisons with those obtained from first- and second-order upwind schemes. Our results show that the RCIP scheme preserves the pattern of input surface mass balance histories in terms of the vertical profile of internal annual layer thickness, better than the other schemes.

## 1 Introduction

Core samples extracted from ice sheets can provide an archive of past climate history data, and a major issue for researchers attempting to utilize ice-core properties is defining the age of ice along the depth of the ice sheet. This process is often called *dating*. Dating with numerical ice-flow models is an important approach because it allows researchers to estimate age profiles before actual drilling of ice cores. For example, in Fischer et al. (2013), the authors present an application of ice-flow models to evaluate potential 'Oldest-Ice' study areas.

Various methods for use in ice-sheet model dating have been adopted and compared. Mügge et al. (1999) compared particle tracking (Lagrangian) and Eulerian schemes under simulated steady-state three-dimensional (3D) velocity fields of Antarctic ice sheet. That study concluded that the Eulerian scheme works well, except for the bottom part, which encounters problems due to numerical diffusion. In Rybak and Huybrechts (2003), the authors also compared the Lagrangian and Eulerian schemes for simulated Antarctic ice sheets under various schematic steady-state conditions and analytical solutions, as well as under different 3D velocity fields. Similarly, they concluded that the Lagrangian method produced less error than an Euler approach, although the difference was small over a large part of the domain. Greve et al. (2002) compared several Eulerian schemes such as central difference schemes, first- and second-order upwind schemes, Quadratic Upstream Interpolation for Convective Kinematics (QUICK), and total variation diminished (TVD) Lax-Wendroff (LW) schemes. From comparisons of the one-dimensional (1D) steady-state age profiles produced by these schemes, they concluded that the second-order upwind and TVD-

LW schemes performed well for typical ice sheet age profiles. Comparisons among semi-Lagrangian schemes have also been performed. Introduction of a semi-Lagrangian trace scheme to ice-sheet modeling was initiated by Clarke and Marshall (2002). They simulate the temporal and spatial variations of water isotopes in the Greenland Ice Sheet over the past 30,000 years. In Tarasov and Peltier (2003), the authors compared various interpolation schemes in order to compute upwind departure points in a semi-Lagrangian tracer model in terms of preservation of input signal phases and amplitudes, while Lhomme et al. (2005); Clarke et al. (2005) developed a new interpolation method that can be used in a semi-Lagrangian scheme and discussed computed ice-core age-depth relationships for the Greenland Ice Core Project (GRIP) ice-core.

To date, various methods have been presented and demonstrated for use in ice-sheet age computations. However, there are still a variety of numerical schemes that have not been examined within this context. These include the constrained interpolation profile (CIP) method (e.g., Yabe et al., 2001) and its variants. Accordingly, the present study introduces a CIP method variant named the rational function-based constrained interpolation profile (RCIP) method (Xiao et al., 1996) for use in 1D ice age computations and demonstrates the performance of the scheme.

## 1.1 Brief introduction of RCIP scheme

This section describes a standard algorithm of the CIP scheme family that is used to solve a 1D advection equation with a non-advection term as follows:

$$\frac{\partial f}{\partial t} + u(x,t)\frac{\partial f}{\partial x} = h(x,t) \,, \tag{1}$$

where $f = f(x,t)$ is a free variable to solve, $u = u(x,t)$ is a velocity field, $h = h(x,t)$ is an arbitrarily (non-advection) field, and $t$ and $x$ are temporal and spatial coordinates, respectively.

As introduced in the previous section, there are three major approaches to solving an advection equation: Eulerian, Lagrangian, and semi-Lagrangian. The CIP scheme family corresponds to a semi-Lagrangian method variation. The basics of the semi-Lagrangian approach, within the context of its comparison with the Lagrangian and Eulerian approaches, have already been presented in a number of past studies. For example, Staniforth and Côté (1990) presented a review of these methods and described the implementation and application of a semi-Lagrangian method in detail. Although a full description of the semi-Lagrangian is not repeated in this paper, its basic principles will be described later in this section.

In CIP schemes, Eq. (1) is solved by performing a time-splitting algorithm (e.g., Yabe and Takei, 1988) into two phases as follows:

$$\frac{\partial f}{\partial t} + u(x,t)\frac{\partial f}{\partial x} = 0 \,, \qquad\qquad \text{the advection phase} \tag{2}$$

$$\frac{\partial f}{\partial t} = h(x,t) \,. \qquad\qquad \text{the non-advection phase} \tag{3}$$

Appendix A presents a note on the time-splitting technique.

The primary characteristic of this CIP scheme is the introduction of an additional equation to solve the spatial derivatives of $f$ at the same time. Differentiation of Eq. (1) provides the equation for $g(x,t) = \frac{\partial f}{\partial x}$:

$$\frac{\partial g}{\partial t} + u(x,t)\frac{\partial g}{\partial x} = \hat{h}(x,t) = \frac{\partial h}{\partial x} - g\frac{\partial u}{\partial x} \,. \tag{4}$$

Equation (4) is an advection formula that is similar to Eq. (1) with the non-advection function $\hat{h}(x,t)$ in the right-hand side, which is solved using a time-splitting procedure similar to those used in Eqs. (2) and (3):

$$\frac{\partial g}{\partial t} + u(x,t)\frac{\partial g}{\partial x} = 0\,, \qquad\qquad \text{advection phase} \tag{5}$$

$$\frac{\partial g}{\partial t} = \hat{h}(x,t)\,. \qquad\qquad \text{non-advection phase} \tag{6}$$

The algorithm used to solve the advection phases (Eqs. 2 and 5), which is a core characteristic of the CIP scheme family, is described here, after which the algorithm used to solve the non-advection phases (Eqs. 3 and 6) will be discussed.

In semi-Lagrangian approaches, a particle at $(x, t+\Delta t)$ originates from the position of the upstream departure point $x_{\mathrm{dep}}$ such that

$$f(x, t+\Delta t) = f(x_{\mathrm{dep}}, t)\,, \tag{7}$$

where

$$x_{\mathrm{dep}} = x + \int_{t+\Delta t}^{t} \mathrm{d}t\, u(x,t)\,. \tag{8}$$

Figure 1 shows a schematic illustration of semi-Lagrangian scheme. The particle at $x_j$ at time $t_m + \Delta t$ originates from a particle at $x_{\mathrm{dep}}$, which is not necessarily on a discretized grid point $x_j$. Therefore, the free variable $f(x)$ must be interpolated using the value on the grid points (represented by color shading in the figure).

The CIP method constructs an interpolation function $F_j(x)$ for the $f(x)$ between two adjacent grid-points $x_j$ and its upwind point $x_{j+1}$ when $u_j < 0$ in order to assess the value at the departure point. Introducing $\langle \xi \rangle = x_{\mathrm{dep}} - x_j$ as the distance to the original point allows the time evolution of $f(x_j)$ (which is the original free variable to solve) and $g(x_j)$ (which is the spatial derivative of $f$ at the grid-points $x_j$) to be solved as simple advection equations:

$$\begin{cases} f(x_j, t+\Delta t) = f(x_j + \langle\xi\rangle, t) = F_j(x_j + \langle\xi\rangle)\,, \\ g(x_j, t+\Delta t) = g(x_j + \langle\xi\rangle, t) = G_j(x_j + \langle\xi\rangle)\,, \end{cases} \tag{9}$$

where $G_j(x) = \frac{\partial F_j}{\partial x}$. Note that computation of distance to the departure point will be described in Sect. 1.2. The piecewise interpolation function $F_j(x)$ for $x_j \le x \le x_{j+1}$ is defined to be constrained by the continuity condition at $x_j$ and $x_{j+1}$ as

$$\begin{cases} F_j(x_j) = f(x_j)\,, \quad F_j(x_{j+1}) = f(x_{j+1})\,, \\ G_j(x_j) = g(x_j)\,, \quad G_j(x_{j+1}) = g(x_{j+1})\,. \end{cases} \tag{10}$$

A cubic polynomial is chosen in the original CIP scheme, as

$$F_j(X) = C_0 + C_1 X + C_2 X^2 + C_3 X^3\,, \tag{11}$$

where $X = x - x_j$. The four coefficients $C_0$, $C_1$, $C_2$, and $C_3$ in Eq. (11) are determined to satisfy the constraints (Eq. 10). The RCIP scheme framework is occasionally extended to introduce a rational function (Xiao et al., 1996) such as

$$F_j(X) = \frac{C_0 + C_1 X + C_2 X^2}{1 + D_1 X}\,. \tag{12}$$

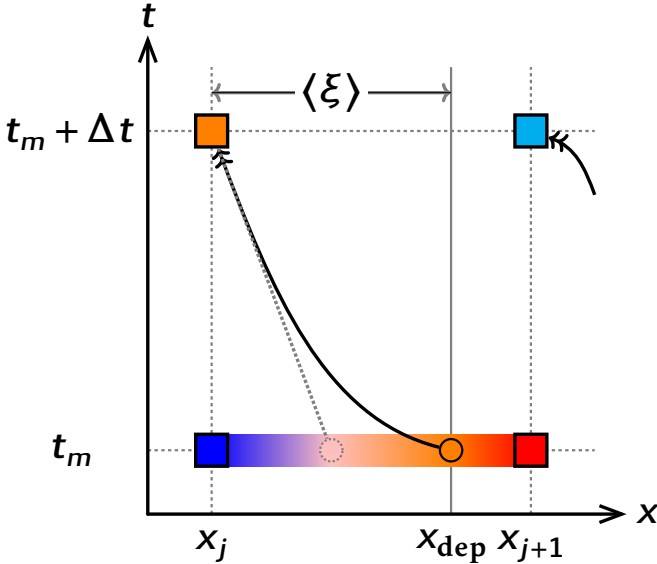

**Figure 1.** Schematic illustration of advection and semi-Lagrangian scheme. The new state computation for a target point $x_j$ from time $t_m$ in the case of $u_j < 0$ is presented. The colors symbolically express the value of field variables. The boxes correspond to model grid points. The solid arrow is the trajectory of one particle and the solid circle is the departure point. In a semi-Lagrangian scheme, the distance to the departure point, $\langle \xi \rangle$, is computed using an assumed trajectory. Interpolating the state at departure point $x_{\mathrm{dep}}$, the value is advected to the arrival point $(x_j, t_m + \Delta t)$. The dotted arrow and circle correspond to the trajectory and departure point in a different case while assuming a constant velocity, which may lead to a different state.

The interpolation function is switched from the cubic (11) to the rational (12) if $g_j \leq S_j \leq g_{j+1}$ or if $g_j \geq S_j \geq g_{j+1}$, where

$$S_j = \frac{f_{j+1} - f_j}{\Delta x_{j+\frac{1}{2}}} , \qquad \Delta x_{j+\frac{1}{2}} = x_{j+1} - x_j . \tag{13}$$

85

Additionally, the four coefficients $C_0$, $C_1$, $C_2$, and $D_1$ are determined in order to satisfy the same constraints. The two interpolation functions Eqs. (11) and (12) are integrated by introducing a switching parameter $\alpha$:

$$F_j(X) = \frac{C_0 + C_1 X + C_2 X^2 + C_3 X^3}{1 + \alpha D_1 X} . \tag{14}$$

The five coefficients used to satisfy the constraints are computed as

$$D_1 = \frac{1}{\Delta x_{j+\frac{1}{2}}} \left[ \left| \frac{S_j - g_j}{g_{j+1} - S_j} \right| - 1 \right] , \tag{15}$$

$$C_3 = \frac{g_j - S_j + (g_{j+1} - S_j)(1 + \alpha D_1 \Delta x_{j+\frac{1}{2}})}{\Delta x_{j+\frac{1}{2}}^2} , \tag{16}$$

$$C_2 = S_j \alpha D_1 + \frac{S_j - g_j}{\Delta x_{j+\frac{1}{2}}} - C_3 \Delta x_{j+\frac{1}{2}} , \tag{17}$$

$$C_1 = g_j + f_j \alpha D_1 , \tag{18}$$

$$C_0 = f_j . \tag{19}$$

The switching parameter $\alpha \in [0, 1]$ is chosen as 1 when it is necessary to use rational interpolation. In other cases, 0 is selected. If $\{f_j\}$ and $\{g_j\}$ at time $t$ are known, the new states $\{f_j^*\}$ and $\{g_j^*\}$ are predicted by shifting by distance along the characteristics (Eq. 9) to the departure point $\langle \xi \rangle$, as follows:

$$\begin{cases} f_j^* = F_j(\langle \xi \rangle) = \dfrac{C_0 + C_1 \langle \xi \rangle + C_2 \langle \xi \rangle^2 + C_3 \langle \xi \rangle^3}{1 + \alpha D_1 \langle \xi \rangle} , \\ g_j^* = G_j(\langle \xi \rangle) = \dfrac{C_1 + 2 C_2 \langle \xi \rangle + 3 C_3 \langle \xi \rangle^2}{1 + \alpha D_1 \langle \xi \rangle} - \dfrac{\alpha D_1}{1 + \alpha D_1 \langle \xi \rangle} f_j^* . \end{cases} \tag{20}$$

The solutions above are those of the advection phases (Eqs. 2 and 5). The time evolutions of $f$ and $g$ in the non-advection phases are again calculated according to Eqs. (3) and (6), typically by using a forwarding scheme, starting from the solution of the advection phase $\{f_j^*\}$ and $\{g_j^*\}$, as an intermediate solution:

$$\begin{cases} \dfrac{f_j(t + \Delta t) - f_j^*}{\Delta t} = h_j \\ \dfrac{g_j(t + \Delta t) - g_j^*}{\Delta t} = \hat{h}_j . \end{cases} \tag{21}$$

As discussed in Xiao et al. (1996), the formulation of the RCIP scheme possesses attractive properties, such as convexity and monotone preservation, as well as phase speed.

## 1.2 Upstream departure point

The interpolation method used for the field variables, which characterize each scheme, is one of the most important topics in semi-Lagrangian schemes. Another major topic common to the semi-Lagrangian schemes is the method used to compute the departure point.

Equation (8) gives the distance to the departure point:

$$\langle \xi \rangle = - \int_{t}^{t + \Delta t} dt\, u(x, t) . \tag{22}$$

A simple and primitive way to integrate Eq. (22) is to use the local velocity even if the velocity is a function of time and space (e.g., Toda et al., 2009), such that

$$\langle \xi \rangle = -u_j \Delta t \ . \tag{23}$$

Figure 1 shows a trajectory and departure point under a constant velocity (dotted line and circle) assumption. As can be seen, the computed departure point can be different from a general non-uniform velocity situation. Another way is to apply the 'mid-point rule', where both spatial and temporal mean velocity between the target and departure points replaces $u_j$ in Eq. (23), which is generally computed in an iterative fashion (Tarasov and Peltier, 2003). In the present paper, a third approach is adopted. First, a steady and linear velocity field between the target and the upstream adjacent points, $x_j$ and $x_{j+1}$, is assumed such that

$$u(x) = u(x_j) + (x - x_j)u' \qquad \text{for } x_j \leq x \leq x_{j+1}, \tag{24}$$

where $u'$ is a constant spatial gradient of the velocity. In order to solve the time evolution of the velocity of a particle at $(t_m, x_j)$, Eq. (24) is differentiated by time $t$:

$$\frac{\mathrm{d}u}{\mathrm{d}t} = \frac{\mathrm{d}x}{\mathrm{d}t}u' = u'u \ , \tag{25}$$

which is solved as

$$u(t) = u(t_m) \exp\left[u'(t - t_m)\right] \ . \tag{26}$$

Introducing Eq. (26), Eq. (22) is integrated as

$$\langle \xi \rangle = -\int_{t_m}^{t_m + \Delta t} \mathrm{d}t \, u(t) = -u(t_m)\Delta t \left[\frac{\exp(u'\Delta t) - 1}{u'\Delta t}\right] , \qquad \text{(when } u' \neq 0) \tag{27}$$

$$\langle \xi \rangle = -u(t_m)\Delta t \ , \qquad \text{(when } u' = 0). \tag{28}$$

Based on the above, it can be interpreted that the distance to the departure point is that of constant velocity case (Eq. 23 or 28), multiplied by the bracket term in Eq. (27) as a correction factor. Here, it should be noted that the correction factor reaches 1 toward the limit of $u' \to 0$, which definitely corresponds to the constant velocity case. The velocity gradient $u'$ already appears in the advection equation of the $g$ term (Eq. 4), which is reused in the departure point computation.

## 2 Model description

### 2.1 Governing equation

The computation used to determine the age of the ice, i.e., the elapsed time since the ice deposit, is performed with the pure advection equation[1]:

$$\frac{\mathrm{d}\mathcal{A}}{\mathrm{d}t} = 1 , \tag{29}$$

where $\mathcal{A}$ is the age and $t$ is time, which is the Lagrangian approach. Eq. (29) is then reformulated into the Eulerian equation for a 1D problem,

$$\frac{\partial \mathcal{A}}{\partial t} + w(z,t)\frac{\partial \mathcal{A}}{\partial z} = 1 , \tag{30}$$

where $\mathcal{A} = \mathcal{A}(z,t)$ and $w = w(z,t)$ are the age and vertical velocity fields, respectively, and $z$ is the vertical coordinate. Some models introduce an artificial diffusion term in order to achieve stable integration(e.g., Mügge et al., 1999). However, the pure advection form is kept throughout the present paper. Following most large-scale numerical ice-sheet models (Greve and Hutter, 1995), the vertical coordinate $z$ is scaled with the local thickness. Introducing the scaled coordinate $\zeta$ as

$$\zeta = \frac{z-b}{H} , \tag{31}$$

Eq. (30) is reformulated as follows:

$$\frac{\partial \mathcal{A}}{\partial \tau} + \omega \frac{\partial \mathcal{A}}{\partial \zeta} = 1 , \tag{32}$$

where $\tau \equiv t$ is the corresponding time coordinate in this system, $b = b(t)$ is the bedrock topography, and $H = H(t)$ is the ice thickness. The new velocity term $\omega = \omega(\zeta, \tau)$ in $\tau, \zeta$-system is computed as

$$\omega = w\frac{\partial \zeta}{\partial z} + \frac{\partial \zeta}{\partial t} , \tag{33}$$

where derivatives of $\zeta$ are computed as:

$$\frac{\partial \zeta}{\partial z} = \frac{1}{H} , \tag{34}$$

$$\frac{\partial \zeta}{\partial t} = -\frac{1}{H}\left[\frac{\partial b}{\partial t} + \zeta\frac{\partial H}{\partial t}\right] . \tag{35}$$

Since the ice thickness $H$, which actually reflects the changes in the boundary conditions, may not be constant throughout the

time period, $H = H(t)$ is prescribed independently of the boundary conditions in this paper. The surface mass balance term $M_{\mathrm{s}}$ (mass input into the domain), surface evolution, and the vertical velocity at the surface $z = h(t)$ are related as

$$w(z = h(t)) = \frac{\partial h}{\partial t} + M_{\mathrm{s}}(t) , \tag{36}$$

---

[1]Some models adopt 0 for the right-hand side (e.g., Rybak and Huybrechts, 2003) simply because they use a different age definition. For such cases, redefining $\mathcal{A}$ as $\mathcal{A} - t$ results in an equation that is identical to Eq. (29).

which is derived from the kinematic boundary conditions based on the assumption of a flat surface. The spatial derivative of $\mathcal{A}$ used in the RCIP scheme is derived as follows:

$$\frac{\partial \mathcal{A}'}{\partial \tau} + \omega \frac{\partial \mathcal{A}'}{\partial \zeta} = -\frac{\partial \omega}{\partial \zeta} \mathcal{A}' \,, \tag{37}$$

where $\mathcal{A}' = \frac{\partial \mathcal{A}}{\partial \zeta}$.

In order to solve the time evolution of age and its gradient (Eqs. 32 and 37), the initial and boundary conditions are required. At the free surface $z = h(t)$ (or $\zeta = 1$), a Dirichlet-type boundary condition,

$$\mathcal{A}(\zeta = 1) = 0 \,, \tag{38}$$

holds when the surface mass balance is positive (i.e., $M_\mathrm{s} > 0$). In contrast, when the surface balance is negative, the boundary condition is not necessary, because the departure point of the free surface is inside the ice. A special treatment is required for the zero mass balance at the surface, $M_\mathrm{s} = 0$. In this case, the velocity term in $\tau, \zeta$-system, $\omega$ becomes 0, so Eq. (32) is simplified as

$$\frac{\partial \mathcal{A}}{\partial \tau} = 1 \,, \tag{39}$$

which, again, requires no boundary condition for age. The boundary conditions at the bottom $\zeta = 0$ simply mirror those at the surface.

The age derivative, $\mathcal{A}'$, also satisfies the boundary condition at the free surface as

$$\mathcal{A}'(\zeta = 1) = -\frac{1}{M_\mathrm{s}} \,, \tag{40}$$

when $M_\mathrm{s} > 0$. Conditions similar to age hold for the age derivative when $M_\mathrm{s} < 0$ and $M_\mathrm{s} = 0$.

In the present study, equivalent but different coefficient representations (Eqs. 15–19) are adopted for the RCIP method implementation, which is described in Appendix B.

## 2.2 Discretization

The spatial discretization of Eqs. (32) and (37) can be either uniform or non-uniform. In the present paper, both types of discretization are examined. Since uniform discretization is a special case of non-uniform discretization, the latter can be described effectively without a loss of generality.

One way to introduce a non-uniform discretization is to apply a *non-smooth* grid (Shashkov, 1995), which prescribes irregular discretization of the coordinates:

$$0 \equiv \zeta_0 < \zeta_1 < \cdots < \zeta_{\mathsf{N}_k - 1} \equiv 1 \,, \tag{41}$$

and

$$\Delta \zeta_{k+1/2} = \zeta_{k+1} - \zeta_k \,. \qquad \text{for } k = 0, \cdots, \mathsf{N}_k - 2 \tag{42}$$

Another way to introduce a non-uniform discretization is to apply a *smooth* grid (Shashkov, 1995), which uses a smooth function to transform the coordinate system. One more coordinate transformation is then performed for a non-uniform smooth-grid system as follows:

$$\frac{\partial \mathcal{A}}{\partial \mathsf{T}} + \mathsf{W}\frac{\partial \mathcal{A}}{\partial \mathsf{Z}} = 1 \,, \tag{43}$$

$$\frac{\partial \mathcal{A}'}{\partial \mathsf{T}} + \mathsf{W}\frac{\partial \mathcal{A}'}{\partial \mathsf{Z}} = -\frac{\partial \mathsf{W}}{\partial \mathsf{Z}}\mathcal{A}' \,, \tag{44}$$

where $\mathsf{T}$ and $\mathsf{Z}$ are the time and vertical coordinates in the new system. A smooth transformation of $\mathsf{Z} = \mathsf{Z}(\zeta)$ or its inverse $\zeta = \zeta(\mathsf{Z})$ is prescribed where necessary. Similarly, a new velocity term $\mathsf{W} = \mathsf{W}(\mathsf{Z}, \mathsf{T})$ in $\mathsf{T}, \mathsf{Z}$-system is computed as

$$\mathsf{W} = \omega\frac{\partial \mathsf{Z}}{\partial \zeta} \,. \tag{45}$$

Equations (43) and (44), which are the target equations to solve, are simply replacements for Eqs. (32) and (37), respectively. The velocity term $\mathsf{W} = \mathsf{W}(\mathsf{T}, \mathsf{Z})$ is prescribed (as will be explained later). The terms $\mathcal{A}$, $\mathsf{W}$, and 1 on the right-hand side correspond to $f$, $u$, and $h$, respectively, in the RCIP scheme framework (Eq. 1). Although it is possible to introduce further non-uniform discretization on the $\mathsf{Z}$-coordinate, in the present paper, only a uniform discretization is examined on the smooth-grid discretization:

$$\mathsf{Z}_k = \frac{k}{\mathsf{N}_k - 1} \,, \quad \text{for } k = 0, \cdots, \mathsf{N}_k - 1. \tag{46}$$

Actually, the discretization of the $\zeta$-coordinate corresponds to the special case of non-uniform smooth discretization with $\mathsf{Z} \equiv \zeta$. Therefore, for both uniform and non-uniform discretization, the scheme will be described hereafter using the $\mathsf{Z}$-coordinate instead of the $\zeta$-coordinate.

## 2.3 Comparing other schemes with RCIP schemes

In the present paper, two numerical schemes, the first- and second-order upwind schemes, are examined in comparison with the RCIP schemes. While there are other numerical schemes suitable for such comparisons, including Lagrangian, other semi-Lagrangian, or even higher-order upwind schemes, these have already been reported in past studies (Mügge et al., 1999; Greve et al., 2002; Rybak and Huybrechts, 2003, Clarke and Marshall, 2002,Tarasov and Peltier, 2003; Clarke et al., 2005). Furthermore, since our study focuses on a demonstration of RCIP schemes in relation to the topic of ice dating, a wide range of comparisons is beyond the scope of this paper.

The 'first-order' upwind scheme in the present paper evaluates the advection term using the velocity at staggered grid points as follows:

$$\mathsf{W}\frac{\partial \mathcal{A}}{\partial \mathsf{Z}}\bigg|_{\mathsf{Z}=\mathsf{Z}_k} \simeq \mathsf{W}_{k+1/2}\frac{\mathcal{A}_{k+1} - \mathcal{A}_k}{\Delta \mathsf{Z}_{k+\frac{1}{2}}} = \mathsf{W}_{k+1/2}\mathsf{A}'_{\mathrm{I}}(\mathsf{Z}_{k+\frac{1}{2}}) \,, \tag{47}$$

when $\mathsf{W}_{k+1/2} < 0$ and $\mathsf{W}_{k-1/2} < 0$. The velocity at staggered grid points is computed by linear interpolation of the two adjacent velocities at normal grid points. Equation (47) corresponds to numerical integration with the midpoint rule if a Dirichlet-type boundary condition is applied on the upper surface (Eq. 38) and the velocity is kept negative throughout. It is especially

notable that, for ice dating at summits, positive (upward) vertical velocity is rarely considered. Therefore, the mid-point rule formulation mentioned above is sufficient for application. On the other hand, a different approach is generally required for a grid point where two velocities at staggered adjacent grid points have opposite signs. In this paper, the velocity term is simply replaced by that at the normal grid point:

$$\left. W\frac{\partial \mathcal{A}}{\partial Z}\right|_{Z=Z_k} \simeq W_k A'_{\mathrm{I}}(Z_{k+\frac{1}{2}}) , \tag{48}$$

where $W_k < 0$, and $W_{k+1/2}$ and $W_{k-1/2}$ have opposite signs.

For the 'second-order' upwind scheme, the derivative of the age term is replaced by the second-order upwind difference formulation as

$$\left. W\frac{\partial \mathcal{A}}{\partial Z}\right|_{Z=Z_k} \simeq W_k A'_{\mathrm{II}}(Z_k) , \tag{49}$$

where

$$A'_{\mathrm{II}}(Z_k) = \frac{(2\Delta Z_{k+\frac{1}{2}} + \Delta Z_{k+\frac{3}{2}})A'_{\mathrm{I}}(Z_{k+\frac{1}{2}}) - \Delta Z_{k+\frac{1}{2}}A'_{\mathrm{I}}(Z_{k+\frac{3}{2}})}{\Delta Z_{k+\frac{1}{2}} + \Delta Z_{k+\frac{3}{2}}} \qquad \text{for } k < N_k - 2, \tag{50}$$

$$A'_{\mathrm{II}}(Z_k) = 2A'_{\mathrm{I}}(Z_k) - \mathcal{A}'_{k+1} , \qquad \text{for } k = N_k - 2 , \tag{51}$$

for the $W_k < 0$ case. The age derivative at the surface is required ($\mathcal{A}'_{k+1}$ in Eq. 51), which is provided as a boundary condition (Eq. 40). For higher-order numerical schemes, the introduction of a slope limiter is a standard method for suppressing the development of oscillations near a discontinuity and/or steep gradients (details are described in Greve et al., 2002). Although it is possible to apply such slope limiters in irregular grids (Murman et al., 2005), an easier approach was adopted instead. Specifically, the formulation is switched back to the first-order scheme when $A'_{\mathrm{I}} > 0 > A'_{\mathrm{II}}$ or $A'_{\mathrm{I}} < 0 < A'_{\mathrm{II}}$. Although this method may be insufficient to stabilize the solution near a strong discontinuity, the implementation of more sophisticated slope limiters is beyond the scope of the present paper.

## 3 Experiment and Results

### 3.1 Experimental design

Following some modeling studies on the dating of deep drilling sites that used simplified 1D vertical ice flow models (e.g., Parrenin et al., 2007), the present study adopts an analytical vertical velocity profile under the assumption that there are no horizontal variations in the bedrock elevation, surface, and basal mass balances:

$$w(\zeta) = -\left[ \left( M_{\mathrm{s}} + M_{\mathrm{b}} - \frac{\partial H}{\partial t} \right) \tilde{w}(\zeta) - M_{\mathrm{b}} \right] , \tag{52}$$

where $M_{\mathrm{s}}$ and $M_{\mathrm{b}}$ are the surface and basal mass balance (positive is input), respectively, $H$ is the ice thickness, and $\tilde{w}(\zeta)$ is the normalized velocity profile. Assuming no basal sliding, $\tilde{w}(\zeta)$ can be approximated by

$$\tilde{w}(\zeta) = 1 - \frac{p+2}{p+1}(1-\zeta) + \frac{1}{p+1}(1-\zeta)^{p+2} , \tag{53}$$

where $p$ is a parameter for the profile (Parrenin et al., 2007). Under the Glen's flow law with a steady-state isotropic ice condition, $p$ is equal to the flow law exponent $n$ (typically $n = 3$). In addition, the RCIP scheme requires the derivative of the velocity, which is computed using the derivative of $\tilde{w}$, as

$$\frac{\partial \tilde{w}}{\partial \zeta} = \frac{p+2}{p+1} \left[ 1 - (1-\zeta)^{p+1} \right] . \tag{54}$$

In addition to the vertical velocity, the time evolutions of the surface and basal mass balances and the ice thickness are required for the age computations. These will be presented in each of the following sections.

The initial conditions for the $\mathcal{A}$ and $\mathcal{A}'$ fields are set to 0 for all our experiments. In these cases, the age derivative $\mathcal{A}'$ is kept 0 under the level at which the age reaches the integration time. Starting from the 0 field, time integration is computed for $2000\,\mathrm{kyr}$ for most of our experiments.

It is worth mentioning that formulations like Eq. (52), which is a function of normalized depth, make it possible to interpret example results for different configurations when appropriate spatial/temporal dimension scaling is used. In this case, the spatial and temporal characteristic scales can be defined, for example, by the ice thickness and the surface mass balance. This means that the age solution under the configuration of 3 and $0\,\mathrm{cm\,yr^{-1}}$ for the surface and basal mass-balance, respectively, has the same normalized shape as that under 30 and $0\,\mathrm{cm\,yr^{-1}}$, by scaling all the time-related terms as $1/10$.

All the computations in our present study were performed on a personal computer (PC) equipped with an Intel Xeon E5-2609 central processing unit (CPU) and compiled with GNU Fortran. Each surface/basal mass balance, ice thickness, and vertical resolution configuration is repeated using four numerical schemes: the RCIP with departure correction (RCIP+corr), the RCIP without correction (RCIP), the second-order upwind scheme (UP-2), and the first-order upwind scheme with mid-point rule (UP-1). Additionally, a first-order scheme without a mid-point rule (UP-1n) is sometimes used. Multiple 1D-column experiments with different boundary configurations using one numerical scheme are examined simultaneously in one run. For example, the mean computational costs for one run (with 28 different configurations) in the case of 129 levels over $200\,\mathrm{kyr}$ are 30, 28, 32, and 34 seconds, using UP-1, UP-2, RCIP, and RCIP+corr, respectively. Those in the case of 513 levels are 338, 296, 364, and 392 seconds, respectively. Details differ among the configurations, and it takes 30 to $40\%$ more time to perform a RCIP+corr run than to perform a UP-2 or UP-1 run.

## 3.2 A verification experiment using uniform velocity

Before performing an experiment under a typical ice sheet configuration, verification of the numerical model used in the present study is presented under further simplified conditions, namely, the constant velocity case. This is easily performed using Eq. (52), in which the parenthesis term equals 0, in other words, by keeping $H$ constant and setting $M_\mathrm{s} \equiv -M_\mathrm{b}$ for arbitrarily $p$.

Figure 2 shows the computed age profile under the uniform velocity of $-15\,\mathrm{cm\,yr^{-1}}$ and $H = 3000\,\mathrm{m}$. Uniform grid spacing of 129 levels is adopted, which corresponds to $\mathsf{Z} \equiv \zeta$ and $\Delta\mathsf{Z} = \Delta\zeta = 1/2^7$ (i.e., $\Delta z = 23.4375\,\mathrm{m}$) using the smooth grid. The time step is set as $100\,\mathrm{yr}$, which corresponds to the Courant–Friedrichs–Lewy (CFL) condition $\sim 0.64$. The vertical age profile

is formulated as

$$\mathcal{A}(z,t) = \min\left[t, \int\limits_{h}^{z} \mathrm{d}z' \frac{1}{w}\right], \tag{55}$$

thus the exact solution for an uniform velocity is

$$\mathcal{A}(z,t) = \min(t, -z/w_c), \tag{56}$$

where $w_c = -15\,\mathrm{cm\,yr^{-1}}$. For completion purposes, the results of the RCIP scheme are plotted in the figure, which is (by definition) identical to those of RCIP+corr scheme. For the steady state, a linear age profile from $0\,\mathrm{yr}$ at the surface and $20\,\mathrm{kyr}$ at the bottom is expected (corresponding to the thick gray line in Fig. 2), which is obtained by all the methods after integration of around $27\,\mathrm{kyr}$ (not shown). In contrast, the transient states are different among the results of the four schemes examined. Figure 3 shows the computed age profile relative to the exact solution, with three different time steps, 100, 50, and $25\,\mathrm{yr}$, for each scheme. The results of RCIP+corr (and thus RCIP) are shown to be less sensitive to the time step than the upwind schemes, which reflects the fact that both the interpolation and the departure point calculation are successful. At $20\,\mathrm{kyr}$, a linear age profile should be obtained, but all four results show ages that are younger than the exact solution, due to numerical diffusion. Additionally, while all of the schemes show relatively good performance for the upper part, the result obtained by the UP-1 scheme deviates the most from the solution. Specifically, it deviates $1\,\mathrm{yr}$ from around $2/3$ of the total depth and reaches almost $1\,\mathrm{kyr}$ at the bottom, which is already visible in Fig. 2. In contrast, the other results deviate from the solution only near the bottom $\sim 9/10$ of the total depth, and reach $\sim 100\,\mathrm{yr}$, or even less, at the bottom. The error at the bottom of UP-1 is 759 to $902\,\mathrm{yr}$ (3.8 to 4.5%), while that of RCIP is 76 to $98\,\mathrm{yr}$ (0.38 to 0.49%), and the best of UP-2 is even better at 7.5 to $154\,\mathrm{yr}$ (0.04 to 0.77%).

## 3.3 Experiment with steady non-uniform vertical velocity

Hereafter, non-uniform velocity experiments are performed using $p = n = 3$ in Eq. (53). First, simple cases with constant surface/basal mass balances, as well as thicknesses that correspond to steady vertical velocity profiles, are shown. Since Eq. (55) cannot be solved using (52), profiles created by numerical integration (the Runge–Kutta scheme) are used as 'benchmark' solutions in this and the following section.

The ice thickness and the accumulation rate chosen in this and the following sections are $\sim 3000\,\mathrm{m}$ and $\sim 3\,\mathrm{cm\,yr^{-1}}$, respectively, which correspond to typical quantities for the East Antarctic Plateau during the glacial period (e.g., Parrenin et al., 2007; Fischer et al., 2013). On the other hand, there are other cases with similar thicknesses and ten-times higher accumulation ($\sim 30\,\mathrm{cm\,yr^{-1}}$), typically in Greenland or West Antarctica (e.g, Clarke and Marshall, 2002). As mentioned in Sect. 3.1, with proper scaling, the results in these sections can be interpreted as results with such higher accumulation rates. This will be examined in the Discussion (Sect. 4).

Two sets of basal melting are presented: no basal melting and $3\,\mathrm{mm\,yr^{-1}}$. The other two parameters are fixed. Surface mass balance is set as $3\,\mathrm{cm\,yr^{-1}}$ and thickness is set as $H = 3000\,\mathrm{m}$. We use a uniform grid spacing of 129 levels ($\Delta z = 23.4375\,\mathrm{m}$), and the time step is set as $100\,\mathrm{yr}$, which is the same configuration used in the previous section.

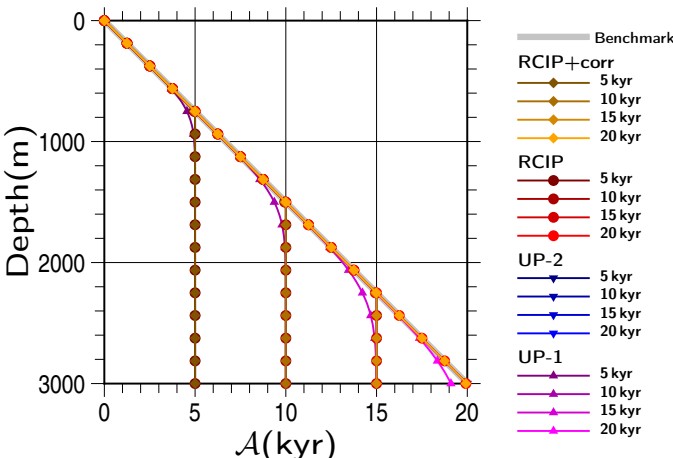

**Figure 2.** Experimental results obtained using a uniform velocity of $w = 15 \, \mathrm{cm \, yr^{-1}}$. Snapshots of the computed vertical age profiles obtained by RCIP with correction, RCIP, second-, and first-order upwind schemes at $t = 5, 10, 15, 20 \, \mathrm{kyr}$ are shown. Since the 'correction factor' of the departure points is 1 (Eq. 28), the results of RCIP with correction are identical to those of RCIP. The results of the second-order upwind scheme are close to those of the RCIP, which are barely visible in this scale. The solution is also shown as a benchmark (thick gray line). Symbols are plotted for every eight vertical levels.

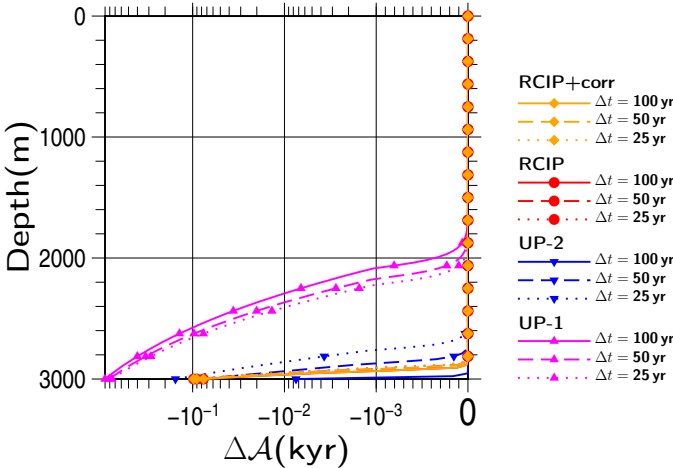

**Figure 3.** Experimental results obtained using a uniform velocity of $w = 15 \, \mathrm{cm \, yr^{-1}}$. Snapshots of computed vertical age profiles obtained by RCIP with correction, RCIP, second-, and first-order upwind schemes at $t = 20 \, \mathrm{kyr}$ relative to the exact solution are shown. The results of different time steps of 100, 50, and 25, yr are shown for each scheme. The results of RCIP with correction are identical to those of RCIP. Age differences are shown on a logarithmic scale, except for those near 0, which are shown on a linear scale. Symbols are plotted for every eight vertical levels.

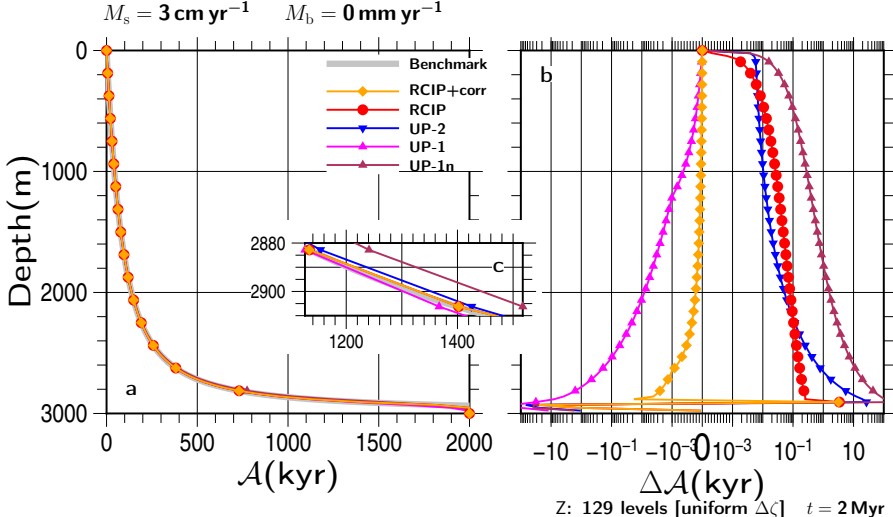

**Figure 4.** Experimental results obtained under steady vertical velocity profiles with $H = 3000\,\mathrm{m}$, $M_{\mathrm{s}} = 3\,\mathrm{cm\,yr^{-1}}$ and $M_{\mathrm{b}} = 0\,\mathrm{mm\,yr^{-1}}$. (a) Computed vertical age profiles by RCIP with departure point correction (RCIP+corr), the RCIP scheme, the second-order upwind scheme (UP-2), the first-order upwind scheme (UP-1), and the first-order upwind scheme without the mid-point rule (UP-1n) at $t = 2000\,\mathrm{kyr}$, (b) those relative to the *benchmark* profile obtained by numerical integration, and (c) a zoomed-in portion of the bottom part of (a) showing that the differences among the experiments is on the order of $10\,\mathrm{kyr}$. Age differences are shown on a logarithmic scale, except for those near 0, which are shown on a linear scale. Uniform grid spacing of 129 levels is adopted in this simulation.

Figures 4a and 5a show computed age profiles at $t = 2000\,\mathrm{kyr}$ for all the schemes along with the *benchmark* age profiles. Very few differences can be seen among the profiles over most parts of the figures under this scale. Deviations from the
benchmark are shown in Figs. 4b and 5b. The results of each scheme show larger errors near the bottom than near the upper part. Some results show sudden increases in the error at certain depths, which correspond to the depths around where the age should reach the time of integration.

The RCIP+corr scheme shows the best result for all depths. The UP-1 scheme shows the second-best result, which is even better than the RCIP scheme around the depth of $2600\,\mathrm{m}$. However, it also shows the largest errors among all the schemes
examined at deeper depths. The good performance of UP-1, in spite of its smallest spatial accuracy, which is attributed to the cancellation of errors due to discretization and numerical diffusion, has already been presented in Greve et al. (2002). The mid-point rule formulation (Eq. 47) also plays a role in the increased accuracy. Due to simple situations, such as the one-direction advection and the constant upper boundary conditions, the age profile computation can be formulated as a vertical integration from top to bottom. This means that the mid-point rule integration actually has second-order accuracy. A *true* first-order
upwind scheme can be applied by using Eq. (48) over the whole domain. In this case, vertical integration from top to bottom corresponds to an Euler integration, which has first-order accuracy. Figures 4b and 5b also contain results obtained using such a scheme, marked as UP-1n. However, as expected, when such a normal-grid velocity is introduced for the advection equation, the results have less accuracy than those of the second-order upwind (UP-2). Furthermore, as shown in Greve et al. (2002), the

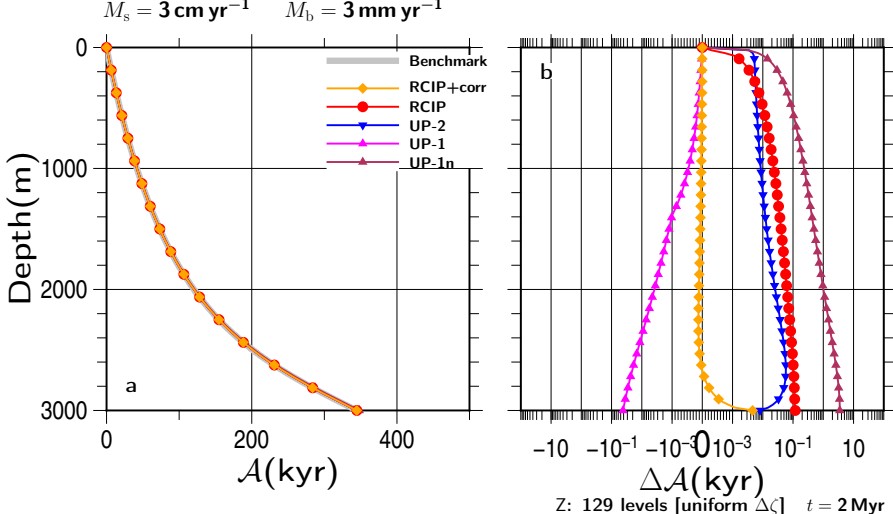

**Figure 5.** Same as Fig. 4a, b, except for the experiment with $M_b = 3 \, \text{mm} \, \text{yr}^{-1}$.

improved performance of the UP-1 scheme is limited to the upper part, and the errors become larger as the depth increases.

The results of the RCIP scheme show relatively larger errors than the other methods, except for the top and the bottom part, which highlights the importance of accurate departure point calculations. The result of the UP-2 scheme shows intermediate errors between RCIP and UP-1 at the bottom.

### 3.4 Non-steady surface mass balance experiments

This section presents the results of experiments conducted with non-steady velocity profiles, which were performed with the

330 prescribed surface mass balance time series. First, a very simple square-wave formulation is adopted for the time evolution of the surface mass balance.

$$
\begin{cases}
M_s(t) = a_H , & (0 \leq \mod(t, P_T) < P_H) \\
M_s(t) = a_L , & (P_H \leq \mod(t, P_T) < P_T = P_H + P_L)
\end{cases} \tag{57}
$$

where $a_H$ and $a_L$ are the prescribed high and low surface mass balance terms, $P_H$ and $P_L$ are the durations with high- and low-value phases, and $P_T$ is the duration of one cycle. Figure 6 shows the time evolution of a normalized surface mass balance

with $P_T = 100 \, \text{kyr}$ cycles and a phase pattern of $P_H, P_L = 1:1$ as an example. Several experimental configuration combinations are examined, including $P_H, P_L = 1:1, 7:1$, or $1:7$; and $M_b = 0, 0.3$, or $3 \, \text{mm} \, \text{yr}^{-1}$. The other patterns examined in this paper are provided as a supplement to this paper.

Figures 7 and 8 show computed age profiles at $t = 1000 \, \text{kyr}$ under the square-wave surface mass balance, where the lower surface mass balances are set as $a_L = 1.5 \, \text{cm} \, \text{yr}^{-1}$ and $0.75 \, \text{cm} \, \text{yr}^{-1}$. The higher surface mass balances and the basal are set

as $a_H = 3 \, \text{cm} \, \text{yr}^{-1}$ and $M_b = 0$, respectively. For reference purposes, the benchmark solutions with constant surface mass balances of $a_H$ and $a_L$ are shown with gray lines. The black line is the benchmark solution with the constant surface mass

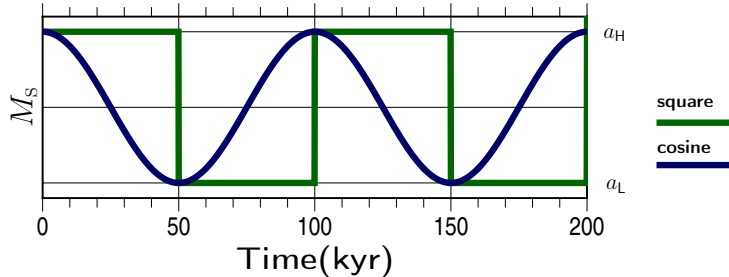

**Figure 6.** Schematic figure showing the time evolution of the surface mass balance adopted in these experiments. Only the first two cycles are plotted.

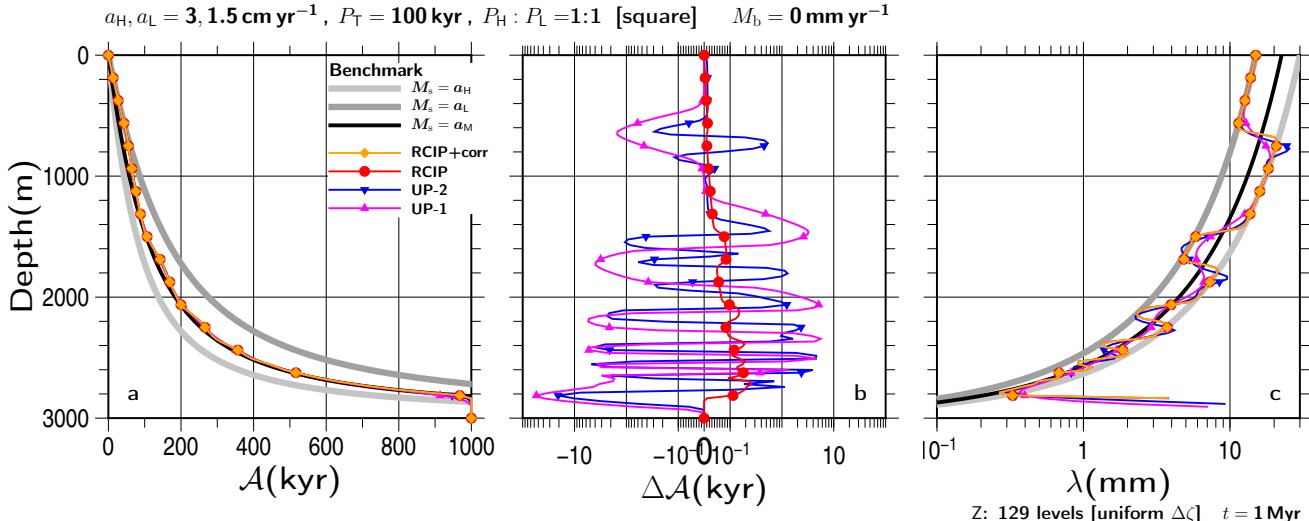

**Figure 7.** Results of transient experiments with square-wave surface mass balances of $a_H, a_L = 3, 1.5\,\mathrm{cm\,yr^{-1}}$, $P_T = 100\,\mathrm{kyr}$, $P_H : P_L = 1 : 1 = 50 : 50\,\mathrm{kyr}$, square-wave, $M_b = 0$, and constant $H = 3000\,\mathrm{m}$. (a) Vertical profiles of the computed age and (c) annual layer thickness at $1000\,\mathrm{kyr}$ using RCIP+corr, RCIP (overlapped on RCIP+corr), UP-2, and UP-1 are shown. The last $500\,\mathrm{yr}$ is clipped from (c), where the age gradient (inverse of $\lambda$) is close to 0 reflecting the initial condition. (b) The computed age differences at the same depth relative to the result of the RCIP+corr case are shown on a logarithmic scale, except for those near 0, which are shown on a linear scale. For reference purposes, the gray lines indicate benchmark solutions for the constant surface mass balance cases of $a_H$, $a_L$, and $a_M$. Uniform grid spacing of 129 levels is adopted in this simulation.

balances of the mean, $a_M = (a_H + a_L)/2$. As shown in these figures, the computed age profiles are close to the benchmark solution with $a_M$, particularly at the bottom. For the upper part, the computed age profiles are along the benchmark solutions for the constant surface mass balance cases of $a_L$.

Since there are few visible differences among the computed ages, the computed age profiles relative to the one produced by the RCIP+corr scheme (Fig. 7b) are shown. The figures show comparable relative performance levels in spite of the different

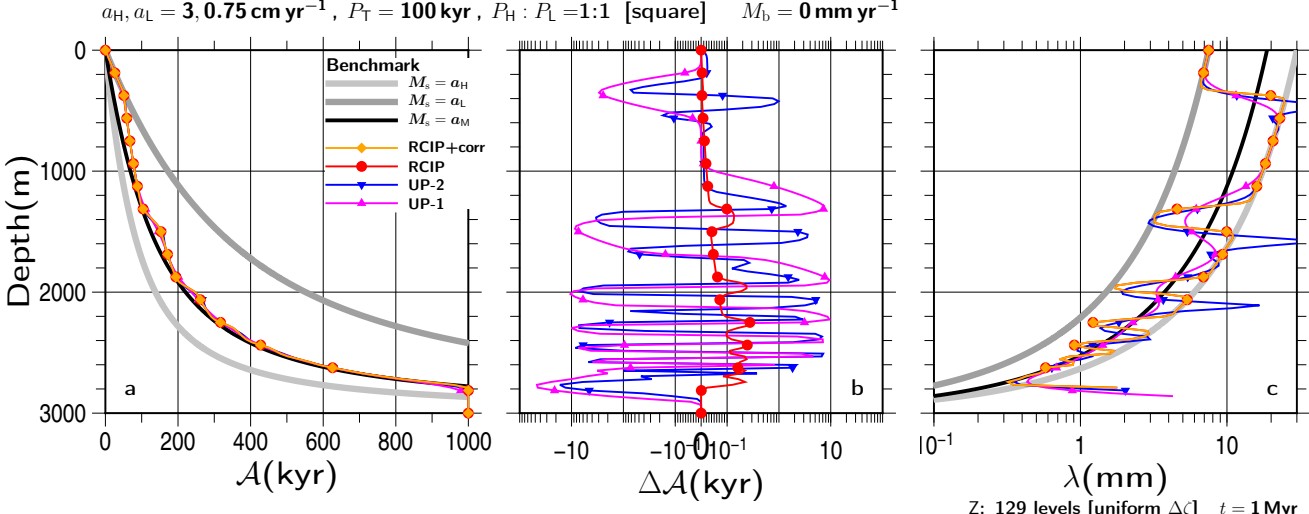

**Figure 8.** Same as Fig. 7, but for the results of transient experiments using a square-wave surface mass balance of $a_L = 3, 0.75\,\mathrm{cm\,yr}^{-1}$.

input surface mass balance histories. The age profiles produced by the RCIP scheme deviate systematically from RCIP+corr by less than $1\,\mathrm{kyr}$ throughout the depth range, which reflects the differences in computing the departure points. The other two schemes deviate around $10\,\mathrm{kyr}$ at most. The age difference oscillations seen in the UP-2 and UP-1 schemes are visible near the age corresponding to the time when switching was conducted between the high and low surface mass balances. ($\Delta\mathcal{A}$ vs. $\mathcal{A}$ plots are presented in the Supplement). These oscillations reflect the characteristics of the UP-2 and UP-1 schemes at the discontinuities.

Figures 7c and 8c show the computed annual layer thickness, $\lambda$, against the depth. In the present paper, the annual layer thickness is defined as the inverse of the age gradient. For the RCIP+corr and RCIP methods, the computed field of the age derivative itself ($\mathcal{A}'$ in Eq. 44) is used with the coordinate transformation. On the other hand, for the UP-2 and UP-1 methods, the diagnosed field is used ($A'_I$ or $A'_{II}$ in Eqs. 47,50,51, respectively). An infinite or a very large annual layer thickness may be present near the bottom, due to the zero gradient of age as a consequence of the initial experiment conditions. In this study, the last $500\,\mathrm{yr}$ is clipped from the figures.

The annual layer thickness has the following relationship in terms of the thinning rate:

$$\frac{\partial\lambda}{\partial t} = \frac{\partial w}{\partial z}\lambda \tag{58}$$

while assuming that layers remain horizontal (Cuffey and Paterson, 2010). When the basal mass balance is $0$ and the thickness is constant, the vertical velocity gradient can be formulated from Eq. (52) as

$$\frac{\partial w}{\partial z} = -M_s\frac{\partial\tilde{w}}{\partial\zeta} \;. \tag{59}$$

Finally after some derivation, the vertical gradient of annual layer thickness can be formulated as

$$\frac{\partial \lambda}{\partial z} = \frac{\lambda}{\tilde{w}} \frac{\partial \tilde{w}}{\partial \zeta} , \tag{60}$$

which is a function of $\lambda$ and the normalized vertical velocity shape. This experiment was conducted with zero basal mass balance, constant thickness, and the same normalized velocity. Therefore, the vertical profile of the annual layer thickness should go back and forth on the two lines produced by those computed using the constant surface mass balances.

In terms of computed annual layer thickness profiles, the RCIP+corr and RCIP (which overlap with RCIP+corr in the figure) methods show particularly good performance over the upper part, as shown in Figs. 7c and 8c. Dissipation at the discontinuity becomes larger towards the bottom, but the solution of RCIP+corr (RCIP) is somewhat more stable on the two benchmark lines than the other schemes. Overshooting at the discontinuity is shown for the solution by the UP-2 scheme, which becomes larger as the difference between the high and low surface mass balances increases. In the present study, this is considered to be a consequence of an inadequate slope filter. In addition, the annual layer thickness is diagnosed with Eq. (50) for the UP-2 scheme, which may exaggerate the oscillation of age gradients more than the simple first-order Taylor expansion. For the UP-1 scheme results, the annual layer thickness diffuses with the depth level and approaches the constant accumulation case of its mean. In deeper areas, the annual layer thickness is found in the vicinity (above or below) of the mean $a_M$ benchmark profile in all of the numerical schemes.

The same exercises were performed using a different shape for the time evolution of the surface mass balance. Figure 9 shows the results for an experiment conducted using the cosine-wave formulation of the surface mass balance (Fig. 6), which is relatively more continuous than the square-wave version. Similar performance levels were obtained by the UP-2 and RCIP+corr (RCIP) methods for the small amplitude case (Fig. 9c). Instability also arises at low-to-high transitions when the ratio of high/low accumulation is larger (archived in the Supplement).

Figure 10 shows the results obtained by square-wave forcing in terms of computed annual layer thickness, $\lambda$, against the computed age for all the schemes, obtained by the relative duration of the $P_H : P_L = 1 : 1$ case (similar figures obtained by other experimental configurations are archived in the Supplement). Since the periodicity of the input cycle is $100 \, \text{kyr}$ in this experiment, the annual layer thickness profiles should show the same periodicity. The obtained results show relatively good performance for the RCIP+corr(RCIP) scheme in terms of the phases when compared with the UP-2 and UP-1 schemes. Dissipation at the discontinuity blur the square-wave shapes, particularly at the deeper part, but the phases are still maintained better by the RCIP+corr (RCIP) scheme than by the UP-2 scheme.

## 3.5   Non-steady thickness experiments

The time evolution of the surface mass balance often involves the evolution of ice thickness as a response. In this section, age computation performance levels under non-steady mass balance and ice thickness conditions are presented. In the present paper, the time evolution of thickness is computed as follows:

$$\frac{\partial H}{\partial t} = -\frac{1}{\tau_H} \left\{ H - H_{\text{ref}}(M_s) \right\} , \tag{61}$$

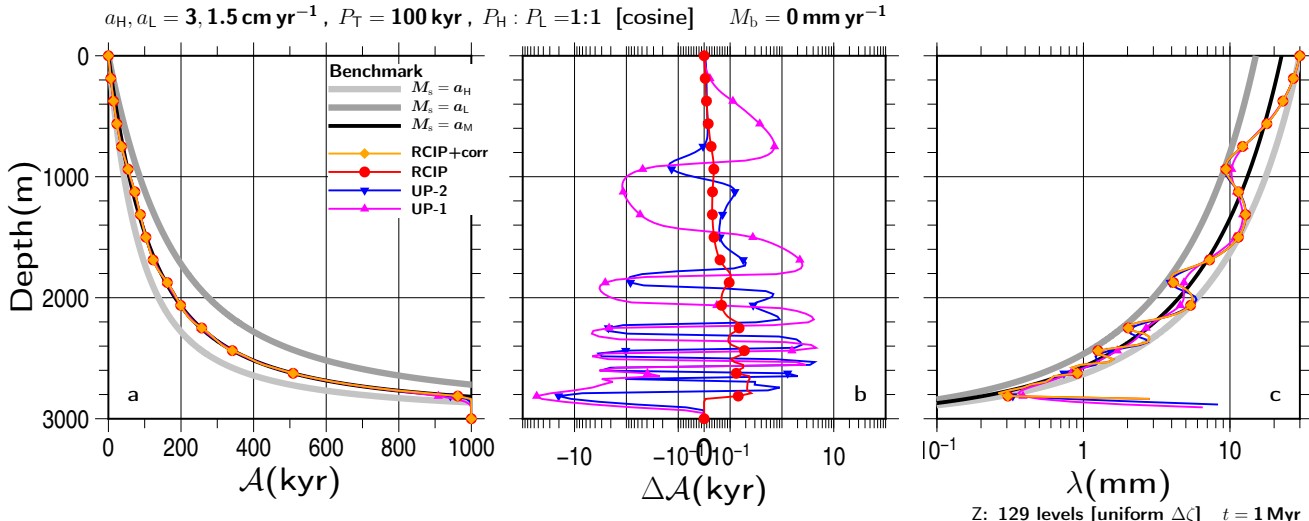

**Figure 9.** Same as Fig. 7, but for the results of transient experiments conducted with a cosine-wave surface mass balance of $a_H, a_L = 3, 1.5\,\mathrm{cm\,yr^{-1}}$.

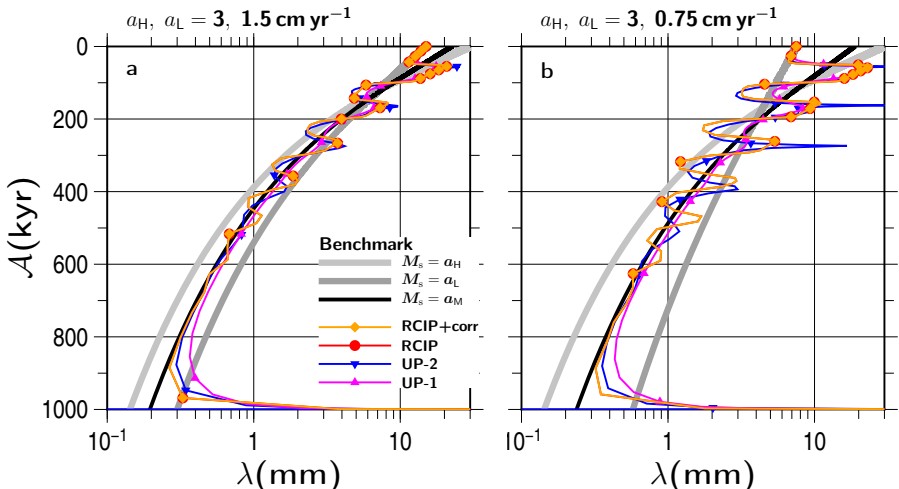

**Figure 10.** Results of transient experiments with square-wave surface mass balances of (a) $a_H, a_L = 3, 1.5\,\mathrm{cm\,yr^{-1}}$ and (b) $3, 0.75\,\mathrm{cm\,yr^{-1}}$ with high- and low-value phase durations (Eq. 57) set as $P_H : P_L = 1 : 1 = 50 : 50\,\mathrm{kyr}$. The basal mass balances are set as 0. The computed annual layer thickness of $\lambda$ against the computing age is shown (RCIP overlaps with RCIP+corr). The gray lines indicate benchmark solutions of the constant surface mass balance cases of $a_H$ and $a_L$, while the black lines $a_M = (a_H + a_L)/2$ are provided as references. Uniform grid spacing of 129 levels is adopted in this simulation.

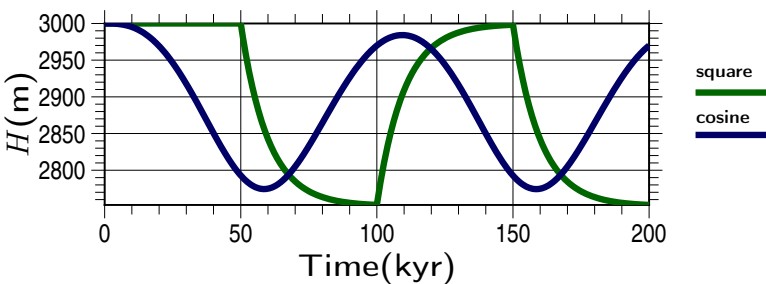

**Figure 11.** Prescribed time evolution patterns of ice thickness adopted in the non-steady thickness experiment. The thickness evolution is computed using an e-folding time of $10\,\text{kyr}$ against square-wave and cosine-wave formulation of the surface mass balance with $a_\text{H}, a_\text{L} = 3, 1.5\,\text{cm}\,\text{yr}^{-1}$ (Fig. 6) provided as an example.

where $\text{H}_\text{ref}(M_\text{s})$ is the reference thickness as a function that depends solely on the surface mass balance and $\tau_H$ is the response thickness timescale. Under ideal conditions, the steady-state ice thickness at the summit is proportional to the $1/(2n+2)$ power of the surface mass balance, where $n$ is the Glen's flow low exponent (Cuffey and Paterson, 2010). Following this relationship, the reference thickness is formulated as

$$400 \quad \text{H}_\text{ref}(M_\text{s}) = H(t=0) \left[ \frac{M_\text{s}(t)}{M_\text{s}(t=0)} \right]^{1/(2n+2)} . \tag{62}$$

For cases where $H(t=0) = 3000\,\text{m}$, $a_\text{H}, a_\text{L} = 3, 1.5\,\text{cm}\,\text{yr}^{-1}$ and $P_\text{T} = 100\,\text{kyr}$, the evolution of $H$ over the first two cycles can be computed as shown in Fig. 11 using Eqs. (61) and 62. The lower thickness limit in this case is $3000 \times (1.5/3.0)^{1/8} \sim 2751.01\,\text{m}$.

Several experimental configuration combinations are examined. These include square-wave or cosine-wave forcing; $a_\text{H}, a_\text{L} = 3, 1.5\,\text{cm}\,\text{yr}^{-1}$ or $3, 0.75\,\text{cm}\,\text{yr}^{-1}$; $\tau_H = 3\,\text{kyr}$ or $10\,\text{kyr}$; $P_\text{H}, P_\text{L} = 1:1$, $7:1$, or $1:7$; $M_\text{b} = 0$, $0.3$, or $3\,\text{mm}\,\text{yr}^{-1}$. Figure 12 shows the result of experiments conducted with response time scales of $10\,\text{kyr}$ for the $100\,\text{kyr}$ cycle square-wave case provided as an example. The gray lines in the figures are the benchmark solution with constant surface mass balances of $a_\text{H}$ and $a_\text{L}$ and their corresponding reference thickness of $\text{H}_\text{ref}$. The black line is computed using the mean surface mass balance $a_\text{M} = (a_\text{H} + a_\text{L})/2$, and the mean thickness over the last cycle of its evolution.

A comparison with the fixed thickness experiments (Fig. 7 vs Fig. 12, or other combinations archived in the Supplement) shows no significant differences. The preservation of discontinuity in the annual layer thickness is similar to that seen in the non-steady thickness case. The differences in computed age, as well as the performance levels of the phases in the annual layer thickness, are qualitatively the same. In addition, all of the combinations examined in this paper show corresponding results that are qualitatively similar to those obtained in the fixed thickness experiments.

### 3.6 Occasional non-positive surface mass balance experiments

So far, the surface mass balance values adopted in our experiments have been positive (corresponding to the accumulation zone). This limitation is sufficient for the usual topics relating to deep ice-core experiments, where the interpretation of ice-

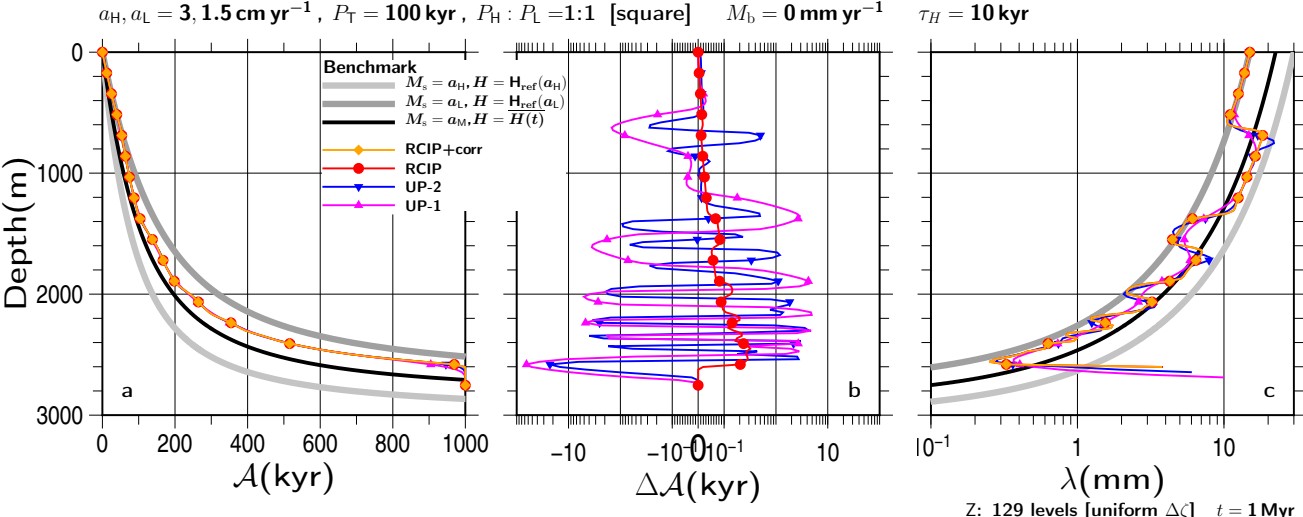

**Figure 12.** Same as Fig. 7, but for the results of non-steady thickness experiments conducted with the response timescale set as $\tau_H = 10$ kyr. The reference thickness values $H$ for the benchmark profiles are explained in the text.

core data may become too complex. However, in order to provide a complete demonstration of the performance levels of numerical age computations for more general cases, it is worthwhile to examine other cases. Although it may be considered

pointless to examine steady negative mass balance cases because they simply mirror the steady positive cases presented above, the surface mass balance level adopted in this section is examined with zero or negative $a_L$ in Eq. (57) and Fig. 6. One encountered difficulty is computing vertical velocity when the surface mass balance is negative. Strictly speaking, it is possible to apply negative $M_s$ to Eqs. (52) and (53), but the validity of such a formulation may be questionable because it is based on an idealized steady-state ice-sheet solution under positive surface mass balance conditions (e.g., Rybak and Huybrechts,

2003). However, for the sake of simplicity, the vertical velocity profiles in the current study are prescribed using the same set of equations for both positive and negative surface mass balances. This is considered to be sufficient, particularly for evaluations of the numerical performance levels of different schemes.

The results of the transient experiments that were conducted under a square-wave surface mass balance of $a_L = 0$ cm yr$^{-1}$ are presented in Fig. 13, while the other configuration is the same as in Figs. 7 and 8. The results obtained under a configuration

with $a_L = -1.5$ cm yr$^{-1}$ are archived in the Supplement. For both experiments, the thickness is fixed as 3000 m, the mass balances are $a_H = 3.0$ cm yr$^{-1}$, $M_b = 0$, and the phases are $P_H, P_L = 50, 50$ kyr.

Several experimental configuration combinations are examined. In a comparison involving the experimental results of the positive mass balance cases examined in this paper, qualitatively similar results are presented. As the prescribed surface mass balance at the lower $a_L$ becomes smaller, errors in the annual layer thickness become clearer around the middle depth. The $\lambda$

for the RCIP+corr(RCIP) scheme at the 1600 m depth and below does not extend to the reference gray line (Fig. 13c). This

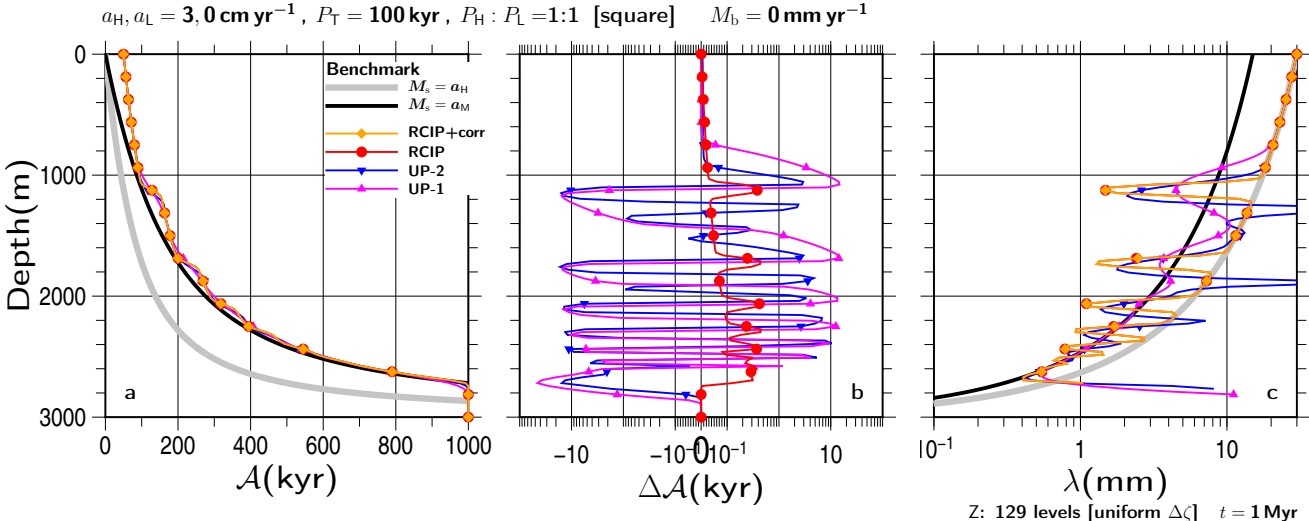

**Figure 13.** Same as Fig. 7, but for the results of transient experiments conducted using a square-wave surface mass balance with $a_{\rm L} = 0\,{\rm cm\,yr^{-1}}$.

is due to the lack of sufficient vertical resolution when capturing the variation. However, the results are still better than those obtained from the other schemes.

### 3.7 Resolution

Annual layer thickness becomes smaller with depth, which reflects the vertical velocity profile. Therefore, differences in age
between two neighboring levels become larger with increasing depth. At a certain depth, the grid spacing becomes insufficient to hold the variation of the input age cycles, which means that the preservation of the input variation is lost below that depth. Typically, in the experiments shown above, $100\,{\rm kyr}$-cycle properties of input surface mass balance are maintained at around $300$ to $500\,{\rm kyr}$ with the RCIP+corr(RCIP) scheme, and the computed age becomes smoother before that age (e.g., the square-wave shape in Fig. 10). The results obtained by UP-2 show loss of variation at similar or shallower depths, and those by UP-1
do so at even shallower depths, which results from the numerical diffusion of the schemes.

In the same manner as computing an approximate depth-age solution under constant surface/basal mass balance and constant thickness (e.g., the gray *benchmark* lines in Fig. 7a), the inverse age-depth solution can be also computed. Using this solution, the vertical profiles of layers that are sufficient to hold a constant age difference ($\mathsf{T}_{\rm res}$) can be obtained. Figure 14b shows four gray lines, which correspond to $\Delta\zeta$ sufficient to hold the 10, 5, 2, and $1\,{\rm kyr}$ differences when the experiment configuration
is constant at $M_{\rm s} = 1.5\,{\rm cm\,yr^{-1}}$, $M_{\rm b} = 0$, and $H = 3000\,{\rm m}$. It is worth mentioning that for other $M_{\rm s}$ cases with the same $M_{\rm b}$ and $H$ constant, the four reference $\zeta$-$\Delta\zeta$ relationships correspond to those with $\mathsf{T}_{\rm res}$ divided by the factor of $M_{\rm s}$ for $M_{\rm s} = 3\,{\rm cm\,yr^{-1}}$, and that they are interpreted as $\mathsf{T}_{\rm res} = 5$, 2.5, 1, and $0.5\,{\rm kyr}$, respectively, while $M_{\rm s} = 0.75\,{\rm cm\,yr^{-1}}$ are interpreted as $\mathsf{T}_{\rm res} = 20$, 10 4, and $2\,{\rm kyr}$, respectively. Therefore, the $\Delta\zeta$ limit estimated by using the lower surface mass

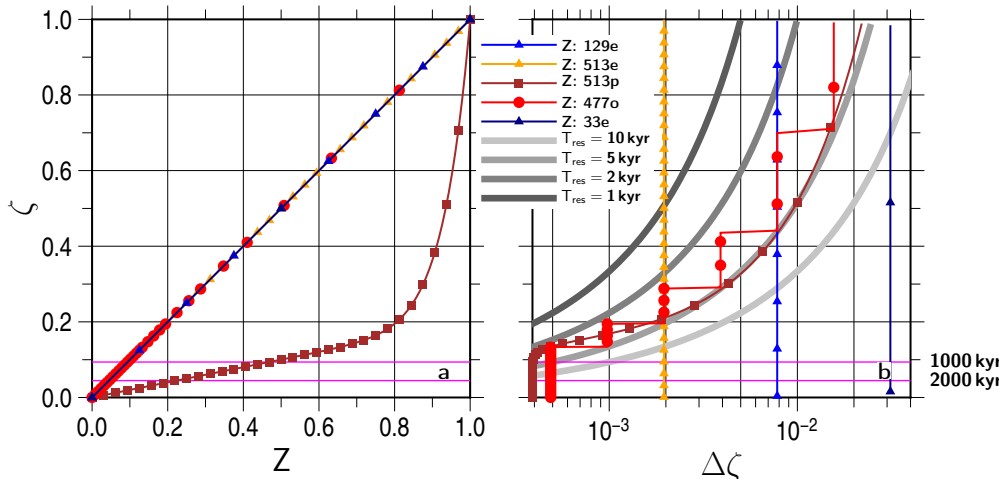

**Figure 14.** Vertical discretization adopted in the present study: (a) Z vs $\zeta$ (b) $\Delta\zeta$ vs $\zeta$. Five patterns are shown: uniform grid spacing of 129 levels (129e), that of 513 levels (513e), a smooth non-uniform discretization (513p, see text), a non-smooth, non-uniform discretization (477o, see text), and uniform grid spacing of 33 levels (33e). Symbols are plotted for every 16 vertical levels. The four gray lines in (b) correspond to the layer thickness necessary to resolve 10, 5, 2, and 1 kyr differences under the condition of $M_s = 1.5\,\mathrm{cm\,yr^{-1}}$, $M_b = 0$, and $H = 3000\,\mathrm{m}$. The two horizontal magenta lines correspond to the depth needed to reach 1000 and 2000 kyr under the same conditions.

balance of the experiment should be considered. For example, if the grid size at a certain $\zeta$ is larger than the $\mathsf{T}_{\mathrm{res}} = 2\,\mathrm{kyr}$ line, characteristics with higher frequencies than $\mathsf{T}_{\mathrm{res}}$ cannot be sampled. The vertical line marked as Z: 129e in Fig. 14b corresponds to uniform grid spacing of 129 levels adopted in the experiments conducted thus far. As the figure shows, this discretization can hold 2, 5, and 10 kyr differences by $\zeta \sim 0.82$, 0.44, and 0.29, respectively, and the 1 kyr is not resolved.

Figure 15 is the same as Fig. 7c, except for the results using the different $P_{\mathsf{T}}$ of 50, 20, and 10 kyr. As shown in the figure, higher-frequency properties disappear at shallower depths. The results of RCIP(RCIP+corr) keep the oscillation relatively stable, but the computed annual layer thicknesses are not on the lines of the constant mass balance cases (gray lines), even at shallow depths, for high-frequency input (Fig. 15c). The square-wave shape pattern seems to be well preserved, at least around the 1700 m depth ($\zeta \sim 0.44$) in Fig. 15a, and around the 600 m depth ($\zeta \sim 0.8$) in Fig. 15b case (which is beyond the range of the figure but presented in the Supplement). Therefore, by comparison with Fig. 14, it can be roughly estimated that $\mathsf{T}_{\mathrm{res}} = 5\,\mathrm{kyr}$ and $\mathsf{T}_{\mathrm{res}} = 2\,\mathrm{kyr}$ or longer are necessary to resolve the $P_{\mathsf{T}} = 50\,\mathrm{kyr}$ and $P_{\mathsf{T}} = 20\,\mathrm{kyr}$ square-wave shapes, respectively, which correspond to $1/10 P_{\mathsf{T}}$.

Here, the same series of experiments is repeated using a higher resolution and a uniform grid spacing of 513 levels, which is four times the resolution of the previous experiments. The vertical line marked as Z: 519e in Fig. 14b corresponds to this grid spacing. As the figure shows, this discretization can hold 5, 2, and 1 kyr differences by $\zeta \sim 0.19$, 0.33, and 0.51, respectively, corresponding to 2430, 2010, and 1470 m in depth, respectively. The time step for higher resolution experiments conducted hereafter is set as 25 yr.

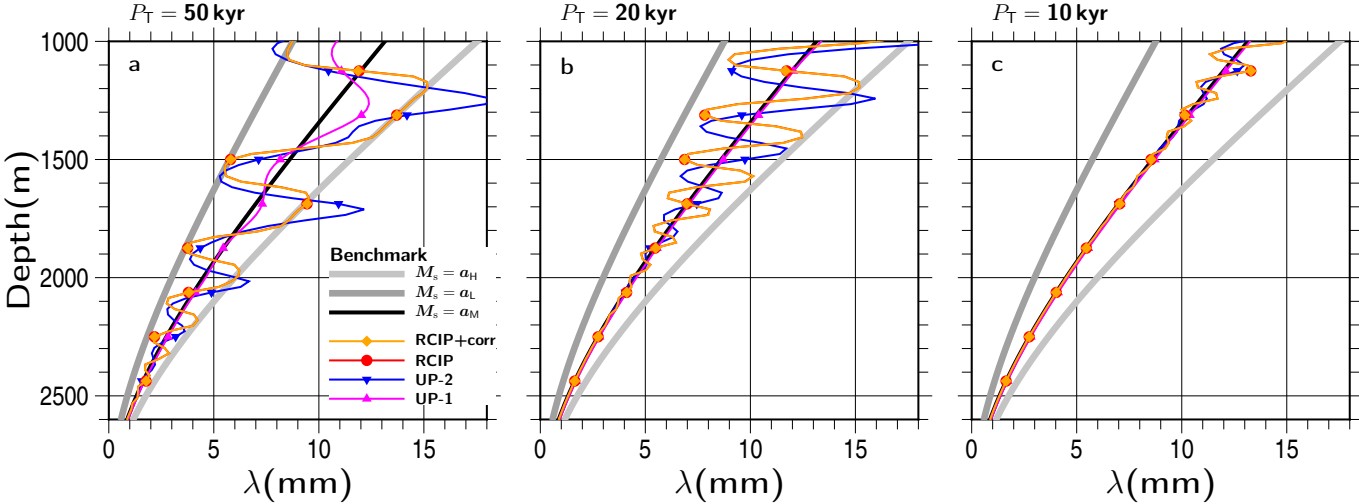

**Figure 15.** Results of transient experiments conducted with the square-wave surface mass balances of $a_H, a_L = 3, 1.5\,\text{cm yr}^{-1}$ and $P_H : P_L = 1 : 1$ (Eq. 57), and the total duration as (a) $P_T = 50\,\text{kyr}$, (b) $P_T = 20\,\text{kyr}$, and (c) $P_T = 10\,\text{kyr}$. The basal mass balance is set as 0 for all the experiments in the figure. The vertical profiles of the annual layer thickness $\lambda$ at $1000\,\text{kyr}$ using RCIP+corr, RCIP (which overlaps with RCIP+corr), UP-2, and UP-1 are shown. The gray and black lines indicate benchmark solutions of constant surface mass balance cases with $a_H$, $a_L$, and $a_M = (a_H + a_L)/2$ given as references. Uniform grid spacing of 129 levels is adopted in this simulation. The results covering the depth from $1000$ to $2600\,\text{m}$ are shown.

Figure 16 is the same as Fig. 15 except for the vertical grid spacing adopted and with zooming shown near the bottom part. The patterns seem to be well preserved by around the depths corresponding to the $\zeta$ above, but the computed ages produced by the RCIP+corr(RCIP) schemes are not on the line $M_s = a_L$ below the corresponding depths.

The number of vertical layers presented above exceeds 100, which is substantially more than those used in many operational
large-scale 3D ice sheet models. The typical number of layers is 30, or even less (e.g., Goelzer et al., 2020; Seroussi et al., 2020). Therefore, since it would be helpful to evaluate performance levels at such a lower resolution for a broader range of applications, a series of experiments was performed using a lower resolution. Figure 17 is the same as Fig. 7, except that the results are provided using a uniform grid spacing of 33 levels (i.e., $\Delta z = 93.75\,\text{m}$), which is one-fourth the resolution of the reference experiment. The time step for the lower resolution experiments was set to $200\,\text{yr}$.

When compared to the higher resolution cases, the annual layer thickness patterns seem to be less preserved. The square-wave pattern in the results of UP-1 has already disappeared at around $1400\,\text{m}$ depth, while those in the other schemes are almost the same, although less than the 129-level cases.

Figure. 14b also contains a line corresponding to a uniform grid spacing of 33 levels marked as Z: 33e. As the figure shows, this discretization can hold $10\,\text{kyr}$ differences by $\zeta \sim 0.67$, corresponding to $990\,\text{m}$ in depth. Similarly, it holds
$20\,\text{kyr}$ differences for $\zeta \sim 0.43$, ($1710\,\text{m}$, not shown in the figure). Figure 17c shows that the result patterns obtained for the RCIP+corr(RCIP) schemes are well preserved by around the $1400\,\text{m}$ depth, which is between the two vertical levels of these

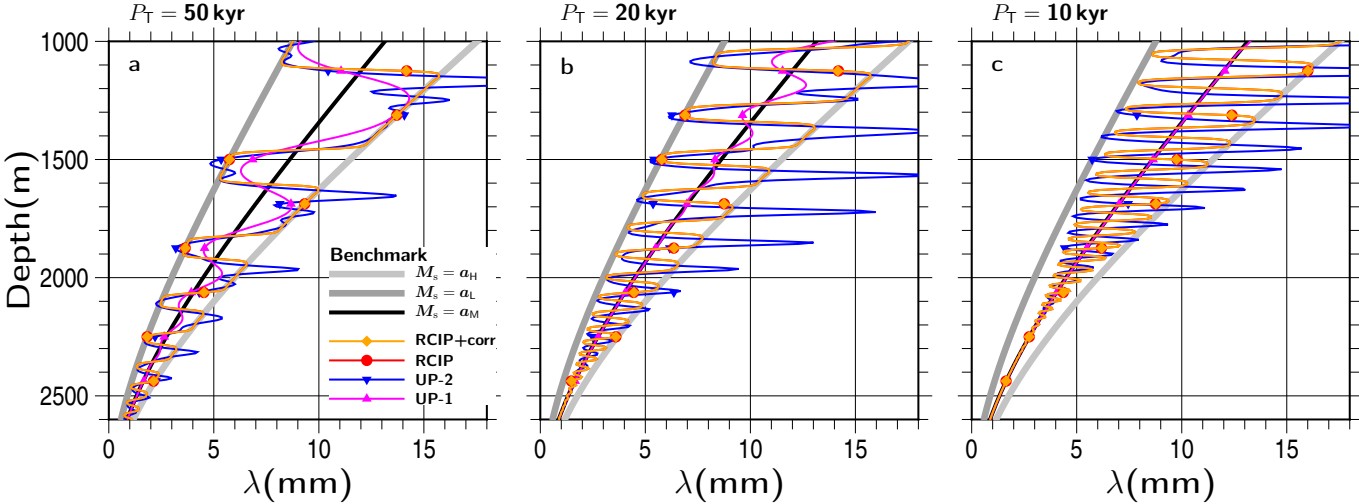

**Figure 16.** Same as Fig. 15, except for the vertical resolution, which is shown as a uniform grid spacing of 513 levels. The results at 2000 kyr covering the depth from 1000 to 2600 m are shown.

estimations. Thus, like the 129-level cases, it can be roughly estimated that $1/10$ (or slightly more) duration of the input cycle is necessary to resolve by one grid.

A comparison between Fig. 17b and Fig. 7b shows that differences in computed age among the schemes are within a comparable range ($\lesssim 10\,\mathrm{kyr}$), and thus are neither exaggerated nor converged by reducing this resolution. The high-frequency oscillation seen near the bottom in Fig. 17b is not in Fig. 17b, which reflects the fact that even the RCIP+corr scheme cannot preserve the input shape near the bottom with the lower resolution. Nevertheless, it should be emphasized that there are still systematic 1 to 10 kyr biases are left by the upwind schemes.

### 3.8 Non-uniform discretization

So far, all of the experiments were performed with uniform discretization of either 129 or 513 levels. For most cases in the present study, it is reasonable to adopt non-uniform discretization, which means large spacing toward the top and small spacing toward the bottom. Since it was previously estimated that at least $1/10$ duration of the input cycle is necessary to resolve one grid, the discretization can be optimized according to the $\Delta\zeta$ profile computed with the minimum surface mass balance of the experiment. Here, for example, the target experiments are set as the square-wave surface mass balance with $P_\mathsf{T} = 100\,\mathrm{kyr}$, $P_\mathsf{H} : P_\mathsf{L} = 1:1$, $M_\mathrm{b} = 0$, $H = 3000\,\mathrm{m}$ constant, and $a_\mathsf{H}, a_\mathsf{L} = 3, 1.5\,\mathrm{cm\,yr^{-1}}$. This is the same configuration seen in Fig. 7. For this configuration, the combination of $\mathsf{T}_\mathrm{res} = P_\mathsf{T}/20 = 5\,\mathrm{kyr}$ and $M_\mathrm{s} = 1.5\,\mathrm{cm\,yr^{-1}}$ is adopted in order to compute the reference profile in the same way as Fig. 14. This number, which is half the number of the estimates used in the discussion above, was chosen for safety, and to facilitate additional experiments with other configurations (e.g., $a_\mathsf{L} = 0.75\,\mathrm{cm\,yr^{-1}}$ and/or $P_\mathsf{T} = 50\,\mathrm{kyr}$, which are archived in the Supplement).

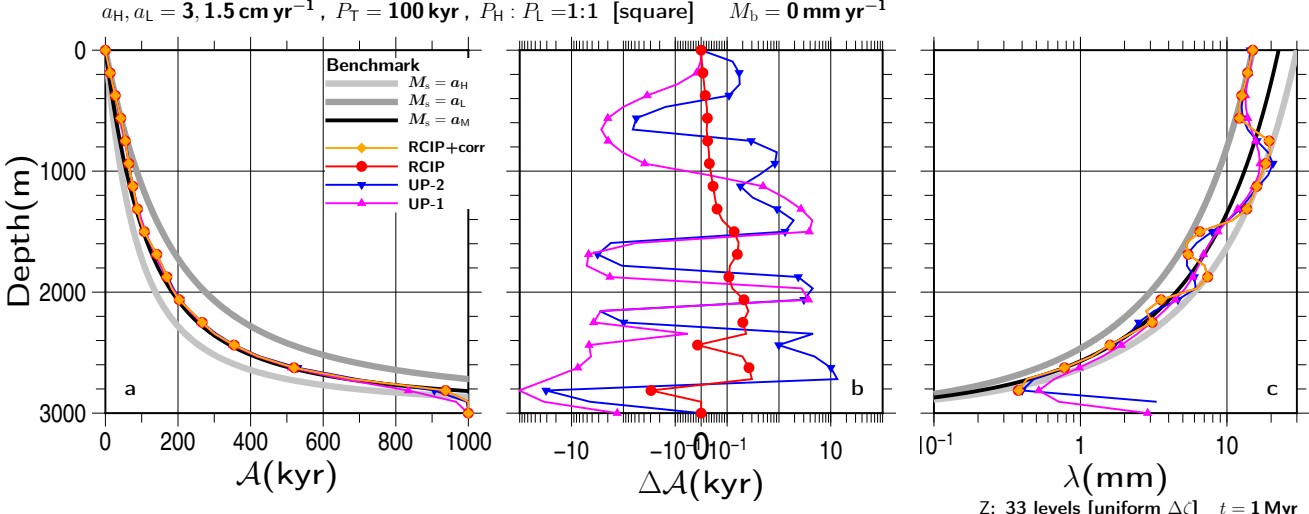

**Figure 17.** Same as Fig. 7, except for the experiment with the lower vertical resolution of 33 uniform grid spacing. Symbols are plotted for every two vertical levels.

Two non-uniform discretization types are adopted in this section. One is a non-smooth grid, and the other is a smooth grid (introduced in Sect. 2.2). Various methods can be applied for non-smooth discretization. A very simple method, which is adopted in this study, calls for starting an initial spacing from the top and then keeping the same grid spacing as long as it is smaller than the reference profile. When the spacing exceeds the reference, it is halved from the coordinates and maintained at that size until it exceeds the reference again. It is necessary to limit the minimum grid spacing in order to avoid an infinite number of discretizations. The line marked as Z: 477o in Fig. 14 is a computed profile following the method mentioned above, which runs from $\Delta\zeta = 2^{-6}$ to $2^{-11}$. It contains 477 levels from $\zeta = 0$ to $\zeta = 1$. The vertical coordinate system Z (model logical coordinate, see Sect.2.2) is identical to $\zeta$, and the series of $\zeta_k$ is shown in Fig. 14a.

For non-uniform smooth discretization, a transformation function that follows the reference profile is necessary between $\zeta$ and Z. Since there is no fixed method for choosing the formulation of Z, the following three constraints are adopted for this paper: (i) $\zeta(Z = 0) = 0$, (ii) $\zeta(Z = 1) = 1$, and (iii) $\frac{d\zeta}{dZ} > 0$ for $0 \leq Z \leq 1$. A simple formulation to satisfy these constraints is

$$\zeta = \frac{Z + \gamma Z^{\psi}}{1 + \gamma}, \tag{63}$$

where the two parameters $\gamma$ (a weight) and $\psi$ (a power) control the shape of transformation. The linear term Z in Eq. (63) is needed to avoid infinite $\frac{\partial Z}{\partial \zeta}$ at $\zeta = Z = 0$, which is used for in Eq. (45). After some trial and error with changing $\gamma$ and $\psi$ chosen from integer numbers, we found the following formulation can be used to maintain a grid spacing that is less than or equal to the reference profile until approaching the bottom:

$$\zeta \equiv \frac{(Z + 4Z^{14})}{5}, \quad Z_k = \frac{k}{N_k - 1} \text{ for } k = 0, \cdots, N_k - 1. \tag{64}$$

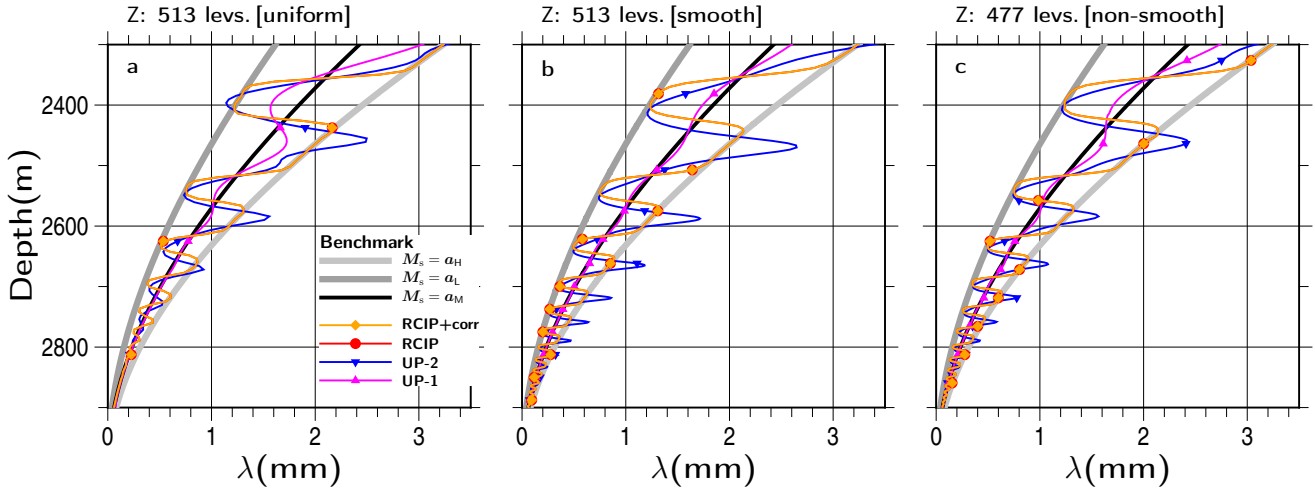

**Figure 18.** Same as Fig. 7c, except for the vertical resolution as (a) uniform grid spacing of 513 levels, (b) smooth non-uniform grid spacing of 513 levels, and (c) non-smooth non-uniform grid spacing of 477 levels. The results at $2000\,\mathrm{kyr}$ covering the depths from 2300 to $2900\,\mathrm{m}$ are shown.

The line marked as Z:513p in Fig. 14 is the profile obtained by (64). Here, the same number of levels (513) is adopted to discretize under the Z-coordinate system. Figure 14a shows the uniform grid spacing of Z, which corresponds to the non-uniform $\zeta$ grid spacing achieved by this method. Additionally, Fig. 14 marks the two vertical coordinates as references in order to reach 1000 and $2000\,\mathrm{kyr}$, respectively, under the constant condition $M_\mathrm{s} = 1.5\,\mathrm{cm\,yr}^{-1}$, $M_\mathrm{b} = 0$, $H = 3000\,\mathrm{m}$.

Figures 18 and 19 show the results obtained using the uniform-spacing (Z:513e), smooth-grid (Z:513p), and non-smooth-grid (Z:477o) discretization methods, in terms of $\lambda$ vs depth, and $\lambda$ vs age. A comparison with Fig. 7 shows that the latter preserves the input shape deeper than the former. As shown in Fig. 14, the uniform discretization case (marked as Z:513e) is expected to fail to resolve $10\,\mathrm{kyr}$ at age $1000\,\mathrm{kyr}$, which is presented in Fig. 19. The results of UP-1 preserved the input shape deeper than the lower resolution, which are almost only half of those achieved with the RCIP+corr(RCIP) methods. The results of UP-2 were preserved at slightly deeper depths than those of UP-1. However, their phases are shown to be shifted from those of the RCIP+corr(RCIP) methods, particularly at the deeper part. For differences in the computed age from the RCIP method (a and b), quantitatively, the same performance levels as the lower-resolution experiment were obtained by the other methods. The UP-1 and UP-2 methods deviate from the RCIP method by around $1\,\mathrm{kyr}$ and $10\,\mathrm{kyr}$ at most, while RCIP+corr deviates by around $100\,\mathrm{yr}$. Using non-uniform discretization, preservation of the input shape is further extended to the deeper part (Fig. 18). As shown in Fig. 14, the non-smooth non-uniform discretization case (marked as Z:477o) crosses the $\mathsf{T}_\mathrm{res} = 10\,\mathrm{kyr}$ line at $\zeta \sim 0.07$ (i.e., $2790\,\mathrm{m}$ depth) for the $a_\mathsf{L} = 1.5\mathrm{cm\,yr}^{-1}$ experiment, which is observed in Fig. 19. The smooth non-uniform discretization case (marked as Z:513p) crosses the $\mathsf{T}_\mathrm{res} = 10\,\mathrm{kyr}$ line slightly below that depth, $\zeta \sim 0.06$ ($2820\,\mathrm{m}$ depth), which is observed in Fig. 19 again. In addition, similar to the lower-resolution experiment, Fig. 19 shows that the RCIP+corr(RCIP) scheme performs relatively better in terms of the phases than those with the UP-2 and UP-1 schemes.

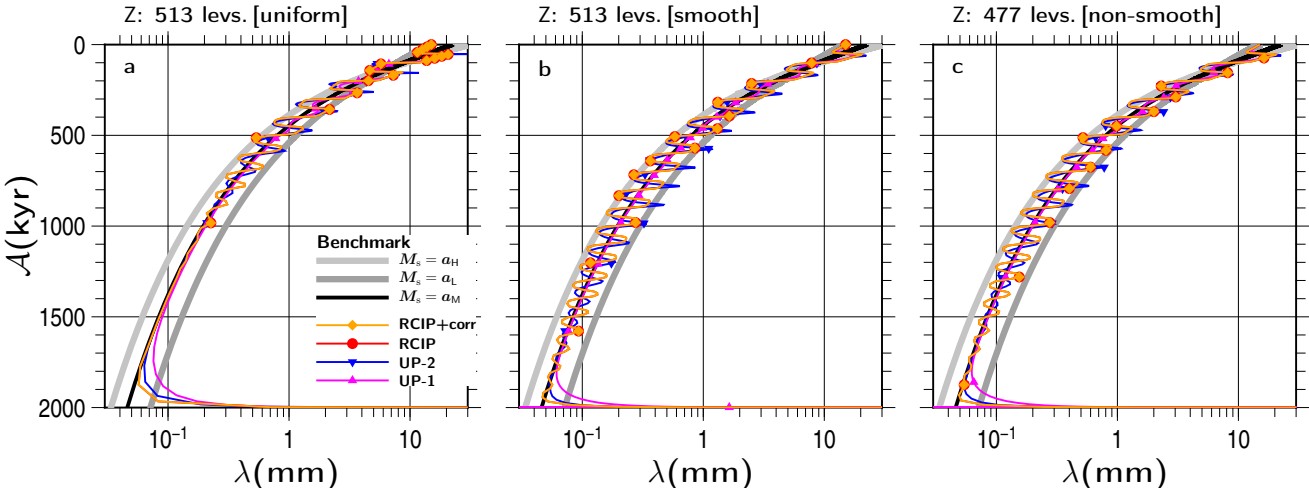

**Figure 19.** Same as Fig. 10, except for the vertical resolution as (a) uniform grid spacing of 513 levels, (b) smooth non-uniform grid spacing of 513 levels, and (c) non-smooth non-uniform grid spacing of 477 levels. The results at 2000 kyr are shown.

## 4 Discussion and conclusion

The present study demonstrates a method for performing 1D age computations of ice-sheets under constant velocity, variable velocity responding to transient changes in surface mass balance, and/or changes in ice-sheet thickness. Herein, comparisons of the vertical profiles of computed ages, as well as annual layer thicknesses, were examined among the RCIP schemes (semi-545 Lagrangian) and upwind schemes (Eulerian). Although the experiments in the present study were limited to 1D computations under summits, we believe the characteristics of the RCIP schemes have been presented sufficiently to allow evaluations of their performance levels.

Overall, the RCIP schemes show the best performance levels among the schemes examined in the present study. In particular, the computed vertical profiles of the annual layer thicknesses produced by RCIP schemes follow the expected depth profiles 550 more reasonably than the other methods. This advantage reflects the design of the RCIP scheme, which explicitly computes the evolution of the age derivative, i.e., the inverse of annual layer thickness, using an advection equation that is similar to the one used to compute the age itself. Using the other schemes, the computed vertical profiles of annual layer thickness either show more smoothing at shallower depths than that were found with the RCIP scheme or the development of oscillation at steep changes in the input surface mass balance. Such oscillation development is shown even when the input is a smooth cosine-555 wave type pattern and the amplitude is large. Since the slope filter adopted in this study is extremely simple, it is possible that the results obtained by the use of a second-order upwind scheme with a more suitable filter will change the characteristics. The introduction of slope limiters on general non-uniform discretization for higher-order upwind schemes is possible (Murman et al., 2005), but the conditions used for switching between a cubic polynomial and a rational form (Eqs. 11 and 12) in the RCIP scheme may be simpler and easier to implement. Under some configurations, oscillation development is not shown by

the second-order upwind scheme. However, the phases of annual layer thickness against the age are shifted from those expected from the initial inputs, which again demonstrates the advantages of the RCIP scheme.

    We examined two methods of computing the departure points in our RCIP scheme experiments. Under a constant velocity case, the results obtained by the simpler method show even less accurate solutions than the first-order upwind scheme, while the other 'correction' method shows the best performance. The computed age differences between the two RCIP methods

is $1000\,\mathrm{yr}$ at most for all the configurations examined in the present study, including the vertical resolution. As a result, the simpler method still performs well if the expected accuracy of the application is less than that period. Under an evolving surface mass balance, the solution of the upwind schemes deviation is by $10\,\mathrm{kyr}$, which is slightly larger.

    As has already been discussed in previous studies(Greve et al., 2002), the first-order upwind scheme shows somewhat better performance than other schemes in some experiments. Greve et al. (2002) attributes this result to the cancellation of errors

between discretization and numerical diffusion. In addition, from comparisons between results obtained via the first-order upwind scheme, with and without the mid-point rule, (Fig. 4), we find that the mid-point rule does provide an advantage because the results obtained without the rule are worse by one order of magnitude than those obtained via the second-order upwind scheme. Furthermore, as discussed above, the upstream correction significantly improves the RCIP solution, which suggests that it is important to consider the non-constant velocity between the arrival and departure points. Since the mid-point

rule formulation in the first-order scheme, in principle, corresponds to this upstream correction, they are consistent. The shape of the normalized vertical velocity profile also may play a role in the relative performance levels. For example, the upper part is more *linear* than the bottom part, which may increase the accuracy of the first-order approximation. In any case, it is clear that some or all of these points contribute to the higher performance of the first-order scheme, except for the bottom part.

    As long as the annual layer thickness is not a concern, we feel that the classical upwind schemes are acceptable choices for

use when dating. Note that using a first-order upwind scheme causes the structural details of the surface mass balance history to disappear very rapidly, but average features will compute quite well, except for near the bottom. The second-order scheme preserves the history better than the first-order scheme, but without an effective slope limiter, strange oscillations can appear in the results, as we have demonstrated in the present paper. However, in spite of these oscillations in the annual layer thickness, the results achieved by the second-order scheme are still slightly better than those for the first-order scheme throughout most

of this study's experiments.

    Greve et al. (2002) presented 'practical suggestions' for numerical dating schemes: the second-order, the total variation diminishing Lax-Friedrichs (TVDLF) scheme with the minimum modulus (min-mod) filter, and even the first-order upwind schemes. In line with those recommendations, we would like to add the following additional practical suggestions. If good performance is required from the annual layer thickness computation, we strongly recommend the application of RCIP. We

also strongly recommend the application of RCIP if age computations near the bottom are required to be within the error range of, e.g., $10\,\mathrm{kyr}$. In other cases, the classical upwind schemes are acceptable choices.

    The ice thickness and accumulation rate values used in the present paper correspond to typical values found on the East Antarctic Plateau, and the values used for the cycles in surface mass balance are 10 to $100\,\mathrm{kyr}$. Providing appropriate scaling is used, all of the results can be interpreted in the same way as those with different configurations. In order to simplify

the situation, the set of calculations will herein adopt a configuration with a different magnitude and surface mass balance cycle, while keeping the same thickness and zero mass balance. Figure 7 is taken as an example. In this experiment, we will use a figure with the same shape while replacing all the time related variables to $1/10$ — ten times higher accumulation rates ($a_\mathsf{H}, a_\mathsf{L} = 30, 15\,\mathrm{cm\,yr^{-1}}$) with $1/10$ cycles, $P_\mathsf{T} = 10\,\mathrm{kyr}$, $P_\mathsf{H}, P_\mathsf{L} = 5, 5\,\mathrm{kyr}$. The range of the horizontal axis needs to be adjusted from (a) $\mathcal{A}$ from 0 to $100\,\mathrm{kyr}$, (b) $\Delta\mathcal{A}$ from $-1$ to $1\,\mathrm{kyr}$, and (c) $\lambda$ from $10^0$ to $10^2\,\mathrm{mm}$, respectively (remember

that the units of *annual* layer thickness $\lambda$ are substantially $\mathrm{mm\,yr^{-1}}$). Similarly, Figure 15 can be interpreted as the results of (a) $P_\mathsf{T} = 5\,\mathrm{kyr}$, (b) $P_\mathsf{T} = 2\,\mathrm{kyr}$ and (c) $P_\mathsf{T} = 1\,\mathrm{kyr}$, respectively, providing that the horizontal axis is adjusted $\lambda$ from $10^0$ to $10^2\,\mathrm{mm}$. In summary, the results in the present paper can be interpreted as cases of $\sim 30\,\mathrm{cm/yr}$ surface mass balance with cycles of 1 to $10\,\mathrm{kyr}$, i.e., millennial-scale climate oscillations on a typical Greenland site. Under the scaled configuration, it can be interpreted from Fig. 7b that the age profiles produced by the RCIP scheme deviate from RCIP+corr by less than $100\,\mathrm{yr}$

throughout the depth range, which reflects the differences in computing the departure points. The other two schemes deviate by around $1\,\mathrm{kyr}$ at most. In addition, from Fig. 15a, it can be seen that the square-wave shape pattern is well preserved, at least around the $1700\,\mathrm{m}$ with $P_\mathsf{T} = 5$ (a) even though the higher-frequency properties disappear at shallower depths with $P_\mathsf{T} = 1\,\mathrm{kyr}$ case (c). Moreover, from examining a series of experiments with ten-times higher accumulation under $1/10$ shorter cycles, we confirmed the same normalized shape of the results (not shown).

Although the focus of the present study is limited to 1D age computations, implementation of the RCIP scheme for 3D computation of the age field is also a suitable subject for future discussions. Extension to 3D would require the consideration of complex 3D flow fields and typically much lower horizontal ice age gradients. In addition, the negative mass balance experiment demonstrated in the present study is too simple to be compatible with the 3D situation. One important characteristic of the CIP scheme family is that the spatial gradient of the field variable (age in this case) is not a diagnostic (passive) value,

but is instead a prognostic field. Yabe et al. (2002) argued that even in an extreme case where values of the three adjacent grid points are zero, one wave still can exist, and thus, non-zero spatial gradients can be held at these grid points. Therefore, it is speculated that the accuracy of the RCIP approach is not worse than that of other semi-Lagrangian schemes using higher-order interpolation techniques over the field variables, which have been discussed in past studies(Clarke and Marshall, 2002; Clarke et al., 2005).

As described in the present study, RCIP is an effective scheme for preserving the flux information at deposition (annual layer thickness in the case of dating). However, detection of 'points of origin' requires another technique, e.g., the back-tracing method. Huybrechts et al. (2007) suggested a very effective back-tracing method, which can be sufficient by itself for ice core dating. The small but primary advantage of the RCIP method over the powerful back-tracing method is that it is a forward scheme. This means that it is not necessary to record all the past velocity field data during the simulation. Therefore we

consider the combination of the high-precision forward scheme and the powerful backward scheme to be a good choice when the objective is to obtain rough and detailed pictures of ice age fields.

Furthermore, it is expected that the RCIP scheme will be applicable to other advection problems in ice sheet modeling. The evolutions of ice-sheet thickness and temperature are formulated using transport or advection equations, which are also good candidates for extending the discussion of this study. For such cases, researchers may be interested in mass or energy conser-

630 vation in the field. Actually, a multi-dimensional conservative formulation of CIP schemes has already been proposed (Yabe et al., 2002). Accordingly, the implementation of the scheme to 3D age and temperature fields in numerical ice-sheet models has already been set as the next target of our development.

*Code availability.* All the numerical experiments in the present paper are performed with IcIES-2/JP version 0, which is a subset package of Ice sheet model for Integrated Earth system Studies II (IcIES-2). IcIES-2 is available at https://github.com/saitofuyuki/icies2 under the
635 Apache license version 2.0. The exact version of the full code, including the scripts used to run the model for all the simulations in this paper, is archived on Zenodo (https://doi.org/10.5281/zenodo.4034557). Also it is tagged as `archive_gmd-2020` in the git repository.

## Appendix A:  Notes on time-splitting

A time splitting technique (Eqs. 2 and 3) is at the core of the CIP schemes, and is somewhat difficult to understand at a first glance. We will attempt to clarify matters with the following simple explanation. If it is assumed that the non-advection term
$h(x,t)$ satisfies at least locally $h(x,t) = h^*(t)$, i.e., is not dependent on $x$, a new variable $f^*(x,t)$ can be introduced such that

$$f^*(x,t) = f(x,t) - \int \mathrm{d}t\, h(x,t) \quad \simeq f(x,t) - \int \mathrm{d}t\, h^*(t)\,. \tag{A1}$$

Then, by introducing Eq. (A1) into the original advection equation (1), a pure advection form of $f^*$ can be obtained,

$$\frac{\partial f^*}{\partial t} + u(x,t)\frac{\partial f^*}{\partial x} = 0\,, \tag{A2}$$

which is the same form as the advection phase equation (2). Using a semi-Lagrangian algorithm, solving Eq. (A2) for $f^*$ at
645 time $t + \Delta t$ requires $f^*(x,t)$, which is identical to $f(x,t)$ by cancellation of the integral term of Eq. (A1). Therefore, Eq. (A2) is solved by the identical procedure used for Eq. (2). After solving $f^*(t + \Delta t)$, $f(t + \Delta t)$ can be computed using Eq. (A1), such that

$$f(x,t+\Delta t) = f^*(x,t+\Delta t) + \int_{t}^{t+\Delta t} \mathrm{d}t\, h^*(t)\,, \tag{A3}$$

which is the same as the non-advection equation (3) and the solution (21) where $f(x)$ is integrated with the initial condition
$f_j^*$.

## Appendix B:  Implementation of RCIP method in the present paper

'Machine epsilon' is defined as the smallest $\epsilon$ in a computer such that $1 + \epsilon > 1$ under floating-point arithmetic. Similarly, an arbitrarily number $f$ has the corresponding smallest number (hereafter $\varepsilon_f$) which satisfies $f + \varepsilon_f > f$. In very rare cases, the authors observed that the age at the upwind grid point becomes close to the value at the target grid point, which differs by $\varepsilon_f$
(i.e., $f_{j+1} = f_j + \varepsilon_f$). Since no representative value exists between $f_{j+1}$ and $f_j$ under floating-point arithmetic, the upwind

value is either $f_j$ or $f_{j+1}$. Sometimes, $f_{j+1}$ corresponds to a value at a grid point that is too far away to be transported. If there is an accumulation of errors of this type, the computed age may show unexpected oscillations.

Although rounding up very small differences may be a possible solution for such cases, a different approach was adopted in the present study. After some trials, the authors adopted finally the following procedure for avoiding such oscillations, which (to the degree they used it) worked better than the rounding-up procedure. In the numerical model of the present paper, Eq. (14) is transformed as follows:

$$F_j(X) = f_j + \frac{g_j X + C_2 X^2 + C_3 X^3}{1 + \alpha D_1 X} = f_j + \delta f , \tag{B1}$$

where $C_1$ and $C_0$ are substituted using Eqs. (18) and (19). The second term $\delta f$ can be computed as the difference between $f_j$ and $f_{j+1}$. When $\delta f$ is non-zero but sufficiently small, i.e., less than $\varepsilon_f$, the value $f_j + \delta f$ is maintained as $f_j$. After simple reformulation, $F_j(X)$ in the model code is finally formulated as

$$F_j(X) = f_j + \frac{\hat{C}_1 X + \hat{C}_2 X^2 + \hat{C}_3 X^3}{\hat{D}_0 + \alpha \hat{D}_1 X} , \tag{B2}$$

where new series of constants are

$$\hat{D}_0 = |g_{j+1} - S_j| , \tag{B3}$$

$$\hat{D}_1 = D_1 |g_{j+1} - S_j| = \frac{|S_j - g_j| - |g_{j+1} - S_j|}{\Delta x_{j+\frac{1}{2}}} , \tag{B4}$$

$$\hat{C}_1 = g_j |g_{j+1} - S_j| , \tag{B5}$$

$$\hat{C}_2 = C_2 |g_{j+1} - S_j| = S_j \alpha \hat{D}_1 + \frac{(S_j - g_j) |g_{j+1} - S_j|}{\Delta x_{j+\frac{1}{2}}} - \hat{C}_3 , \tag{B6}$$

$$\hat{C}_3 = C_3 |g_{j+1} - S_j| = \frac{|g_{j+1} - S_j|}{\Delta x_{j+\frac{1}{2}}} \left[ g_j - S_j + (g_{j+1} - S_j) + \alpha \hat{D}_1 \Delta x_{j+\frac{1}{2}} \right] , \tag{B7}$$

$$\tag{B8}$$

respectively. When $g_{j+1} - S_j = 0$ (and $\alpha = 1$), the coefficients lead to

$$\hat{C}_3 = 0, \quad \hat{C}_1 = 0, \quad \hat{D}_0 = 0 , \tag{B9}$$

$$\hat{C}_2 = S_j \frac{|S_j - g_j|}{\Delta x_{j+\frac{1}{2}}} , \tag{B10}$$

$$\hat{D}_1 = \frac{|S_j - g_j|}{\Delta x_{j+\frac{1}{2}}} , \tag{B11}$$

$$\tag{B12}$$

respectively, and using this combination, $F_j(X)$ is formulated as

$$F_j(X) = f_j + \frac{\hat{C}_2 X^2}{\hat{D}_1 X} = f_j + S_j X , \tag{B13}$$

which means a linear profile is adopted, regardless of $g_j$.

*Author contributions.* FS developed the ice-sheet model and then implemented the RCIP and other dating schemes in the model. FS performed numerical experiments designed by all the authors. The manuscript was written by FS with contributions from TO and AAO.

*Competing interests.* The authors declare that they have no conflicts of interest.

*Acknowledgements.* We would like to thank Shawn Marshall and an anonymous referee for their valuable comments, which have substantially improved our manuscript. This study was supported by the Japan Society for the Promotion of Science (JSPS) KAKENHI under Grant Numbers 17K05664, 17H06323, and 17H06104.

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
