# Peer review of "Implementation of RCIP scheme and its performance for 1D age computations in ice-sheet"

_Geoscientific Model Development, 2020_

## Referee Comment (RC1) · Anonymous Referee #1 · 12 May 2020

**Summary**

F. Saito et al. paper addresses the problem of numerical computation of ice age in ice sheet models. Indeed, calculation of ice age is the major challenge of ice sheet modeling in various applications beginning with the preliminary choice of potential target place for deep drilling of ice sheets and ending with the accurate interpretation of ice cores.

The study area in Saito et al. manuscript is limited by a summit position of an ice sheet where the benchmark – an analytical solution for the ice age can be set. The authors examine two semi-Lagrangian RCIP schemes performance and compare results with more traditional Eulerian upwinding schemes for solving an advection equation.

**General remarks**

It should be noted that a family of RCIP schemes have been applied earlier in various problems of hydrodynamics, hydraulics etc., but their application for ice age calculation in ice sheet modeling is a novel and, perhaps, a promising approach. In reality, of course, we face with 3D problems of ice age computation, either when it is necessary to build the ice age field of the whole ice sheet or to construct a model chronology of a virtual ice core. From this point of view, the submitted research of Saito et al. may be considered as just an academic exercise comparing various numerical methods for highly idealized environmental conditions, which never occur in reality. Nevertheless, such kind of research are useful because they indicate possible pitfalls of rather traditional methods and introduce new approaches for solving tantalizing tasks in ice sheet modeling. In the 'Discussion and conclusion' section the authors reasonably point that the advection problem can be attributed not only to ice age calculation but also to calculation, for instance, of ice temperature. Anyway, further application of the RCIP method and its comparison with the Eulerian schemes in 3D will inevitably face with the choice of a true benchmark, which are, indeed, absent except in case of visual calculation of annual ice layers in ice cores. Moreover, for model interpretation of ice cores a very effective back-tracing method was suggested (Huybrechts et al., Climate of the Past, 2007) which is a powerful tool for dating of ice cores using ice sheet modeling technique. In the 'Discussion and conclusion' section authors mention that their aim is to proceed with examination of the RCIP scheme in 3D. In this view, I think it would be reasonable to outline possible restrictions, challenges and limitations of future research.

Another problem, which was not elucidated in the manuscript is the computational cost of application of different numerical schemes. I think it would be easy to do since all experiments were performed on the same computer facility. There is only short note on that (Line 239). The trade off between time of computation and accuracy in some cases may play for the simpler but faster method.

In general, the manuscript is well structured, the figures are informative (except the note below concerning an a possible additional figure).

**Line by line comments**

The title of the paper. The core of the paper is a set of comparisons between performance of the semi-Lagrangian RCIP schemes and the Eulerian once. Actually, there is nothing in the manuscript about ice sheet models. Therefore, it would be reasonable to be more precise in formulation of the title.

Line 15. "... more generally in tracer transport ...". This statement is somewhat confusing. Dating of ice cores is not limited to tracer transport. This definition (tracer transport) may be attributed to Lagrangian or semi-Lagrangian methods only.

Section 1.1 and 1.2 The section lacks short general description of the semi-Lagrangian method in the context of its comparison with the pure Lagrangian and the Eulerian. Since the problem of interpolation is the most important in semi-Lagrangian schemes, it will be very much handful to make a (sketch) figure illustrating application of a 1-D semi-Lagrangian approach using definitions of the variables mentioned in the manuscript (arrival and departure points etc.). It would be also appropriate to address the reader to a classical paper (Stanoforth and Côté, 1991, Semi-Lagrangian integration schemes for atmospheric models: a review. Mon.Weather Rev., 119(9), 2206-2223.)

Line 61. Please comment on the first use of  $g(x_j)$ . What is it, what is the purpose of its introduction etc.

Line 187. To be precise, Rybak and Huybrechts (2003) did not employ semi-Lagrangian approach, but pure Lagrangian particle tracing.

Line 394. "Figure 14 is the result ..." should be reformulated like, for instance, "Results of transient experiments are presented in Figure 14 ...". Same is in Line 395: "same as IN Fig. 6 AND 7." Same is in the next sentence.

Line 459. Please, check equation for  $\zeta$ . What is Z14? Please, explain why did you use this particular formula for the smooth discretization? What did you mean under "some trial and error". In my view, you should be more exact.

Line 482. Please, indicate that your computations can be related to the summit points of ice sheets only, which are accepted stable throughout the time spell of numerical experiments.

Line 482. In my view, the fragment of the text "... ice-sheets under various configurations" is somewhat confusing. The results of the study are attributed to summits of ice sheets only, and their configurations have no any connection with the research.

---

## Referee Comment (RC2) · Shawn Marshall (Referee) · 23 May 2020

The authors present a detailed examination of a novel (in ice sheets) interpolation scheme with promise for improved tracing of ice age as well as annual layer thickness reconstructions in ice sheets. This study focuses on 1D examples with scenarios (e.g. mass balance accumulation rates/vertical velocities) typical of the East Antarctic plateau, with direct relevance to ice core dating and age modelling.

The study is comprehensive, with superb attention to detail and to explaining the method and the mathematical implementation, such that this should provide a strong foundation for building on and for others that choose to adopt these methods. It is a

valuable study, as age modelling or other passive tracer advection studies (e.g., isotopes, dust layers, or other chemical horizons) have not been given much attention in ice sheet studies in recent years, and are likely due for a resurgence as radar reconstructions are giving increasing detail on 3D ice sheet structure (e.g. McGregor et al., 2015); 3D tracer modelling offers an important avenue for improving and constraining ice sheet models. The methods introduced here should be seriously considered as an alternative to more 'classical' semi-Lagrangian interpolation schemes such as upwind differencing.

MacGregor, J. A., M. A. Fahnestock, G. A. Catania, J. D. Paden, S. Prasad Gogineni, S. K. Young, S. C. Rybarski, A. N. Mabrey, B. M. Wagman, and M. Morlighem (2015a), Radiostratigraphy and age structure of the Greenland Ice Sheet, J. Geophys. Res. Earth Surf., 120, 212–241, doi:10.1002/2014JF003215.

I am attaching a copy of the manuscript with several minor points. The English needs a bit of a double check throughout, for articles, but it is extremely well written and thorough, overall. I will confess that I did not work through the mathematical derivations carefully and have no experience with the RCIP or CIP techniques, so I cannot comment specifically on the rigour and appropriateness of this aspect of the manuscript, or on the novelty of the ideas (vs. e.g., existing implementations in other contexts such as atmospheric models). It is new and relevant to ice sheet modelling.

There is a large number of figures, and it could be worthwhile to consider condensing the presentation of results a little. For instance, with new experiments/sensitivity tests after Figure 7, it could be possible to show only one result (of Figures 8 and 9, and of Figures 14 and 15; maybe elsewhere), while still discussing both experiments in the text. I am also OK with the manuscript as is. Sometimes it is nice to see everything laid out and presented, without relegating additional results to supplements.

A couple of suggestions for the authors' consideration:

The accumulation rates in the experiments are very low, typical of the East Antarctic

Plateau during the glacial period. I guess that it does not affect the performance of the different interpolation/advection models, but am curious to confirm this for the case of e.g. accumulation rates 10 times higher, more typical of Greenland. Also, combined with this, high-amplitude, millennial-scale climate oscillations that are typical of Greenland (D-O cycles). Are there specific recommendations or differences in RCIP behaviour specific to these conditions?

The model is developed specific to 1D age modelling in ice core settings (i.e. purely vertical flow, positive surface mass balance). Extension to 3D is discussed near the end, but would require consideration of positive (emergence) velocities, 3D flow fields, and (typically) much lower horizontal gradients of ice age. This first comes up on p.8, l.197, where the authors develop a formulation that assumes negative vertical velocity throughout, which will not be compatible with 3D modelling. I appreciate that the extension to 3D is for future study and we already have much to chew on with the current presentation of ideas and results, but this discussion could be extended a bit and I am curious about the author's opinion of whether the more complex RCIP type of approach is warranted for the lower horizontal gradients in 3D interpolation models.

Related to 3D models: the authors explore what would be considered as high vertical resolution in ice sheet models, from 129 to 513 vertical layers. This is much higher than many operational 3D ice sheet models that look at 3d (Stokes) solutions to the velocity field or Ice Age timescales: nz = 40 may be more typical. In the section on vertical resolution, it would be helpful to include an experiment with e.g. nz=33 to evaluation model performance at lower resolution. Does it further degrade the interpolation schemes and exaggerate the differences in modelled ice age, or do models converge as resolution declines?

I am interested in the relatively strong results of the first-order upwind scheme. The authors do discuss this, but why is this consistently better than 2nd-order upwind schemes in almost all of the model experiments? In some cases it is of comparable performance to RCIP. Would the authors recommend always using 1st-order over 2nd-order

upstream advection/interpolation models, and under what conditions might 1st-order advection schemes be adequate, vs. the RCIP-corr approach? A short discussion of 'practical suggestions' for eventual application of this technique in ice sheet models would be valuable.

Many thanks for this interesting contribution - I look forward to seeing the final version advance to GMD and push the research community forward.

Please also note the supplement to this comment:
https://www.geosci-model-dev-discuss.net/gmd-2020-53/gmd-2020-53-RC2-supplement.pdf

[Figure]

**Supplement:**

[revised manuscript text omitted]

where $\mathrm{H}_{\mathrm{ref}}(M_{\mathrm{s}})$ is the reference thickness as a function that depends solely on the surface mass balance and $\tau_H$ is the response thickness timescale. Under an idealized condition, the steady-state ice thickness at the summit is proportional to the $1/(2n+2)$ power of the surface mass balance, where $n$ is the Glen's flow low exponent (Cuffey and Paterson, 2010). Following this
365 relationship, the reference thickness is formulated as

$$\mathrm{H}_{\mathrm{ref}}(M_{\mathrm{s}}) = H(t=0)\left[\frac{M_{\mathrm{s}}(t)}{M_{\mathrm{s}}(t=0)}\right]^{1/(2n+2)}. \tag{62}$$

For cases where $H(t=0) = 3000\,\mathrm{m}$, $a_{\mathrm{H}}, a_{\mathrm{L}} = 3, 1.5\,\mathrm{cm}\,\mathrm{yr}^{-1}$ and $P_{\mathrm{T}} = 100\,\mathrm{kyr}$, the evolution of $H$ over the first two cycles can be computed as shown in Fig. 11 using Eqs. (61) and 62. The lower thickness limit in this case is $3000 \times (1.5/3.0)^{1/8} \sim 2751.01\,\mathrm{m}$.

[Figure]

[Figure]

**Figure 11.** Prescribed time evolution patterns of ice thickness adopted in the non-steady thickness experiment. The evolution of thickness is computed using the e-folding time of $10\,\mathrm{kyr}$ against square-wave and cosine-wave formulation of the surface mass balance with $a_{\mathrm{H}}, a_{\mathrm{L}} = 3, 1.5\,\mathrm{cm\,yr^{-1}}$ (Fig. 5) provided as an example.

[Figure]

**Figure 12.** Same as Fig. 6, but for the results of non-steady thickness experiments conducted with the response timescale set as $\tau_H = 10\,\mathrm{kyr}$. The reference thickness values $H$ for the benchmark profiles are explained in the text.

370    Several experimental configuration combinations are examined. These include square-wave or cosine-wave forcing; $a_{\mathrm{H}}, a_{\mathrm{L}} = 3, 1.5\,\mathrm{cm\,yr^{-1}}$ or $3, 0.75\,\mathrm{cm\,yr^{-1}}$; $\tau_H = 3\,\mathrm{kyr}$ or $10\,\mathrm{kyr}$; $P_{\mathrm{H}}, P_{\mathrm{L}} = 1 : 1$, $7 : 1$, or $1 : 7$; $M_{\mathrm{b}} = 0$, $0.3$, or $3\,\mathrm{mm\,yr^{-1}}$. Figures 12 and 13 are the results of experiments conducted with response time scales of $10\,\mathrm{kyr}$ under $100\,\mathrm{kyr}$ cycle square-wave and with cosine-wave forcing cases provided as examples. The gray lines in the figures are the benchmark solution with constant surface mass balances of $a_{\mathrm{H}}$ and $a_{\mathrm{L}}$ and their corresponding reference thickness of $\mathrm{H_{ref}}$. The black line is computed using the mean

375    surface mass balance $a_{\mathrm{M}} = (a_{\mathrm{H}} + a_{\mathrm{L}})/2$, and the mean thickness over the last cycle of its evolution.

    A comparison with the fixed thickness experiments (Figs. 6 and 8 vs Figs. 12 and 13) show no significant differences. The preservation of discontinuity in the annual layer thickness is similar to that seen in the non-steady thickness case. The differences in computed age as well as the performance levels of the phases in the annual layer thickness are qualitatively the

[Figure]

[Figure]

[Figure]

**Figure 13.** Same as Fig. 8, but for the results of non-steady thickness experiments conducted with the response timescale set as $\tau_H = 10\,\mathrm{kyr}$. The reference thickness values $H$ for the benchmark profiles are explained in the text.

same. In addition, all of the combinations examined in this paper show corresponding results that are qualitatively similar to

380 those obtained in the fixed thickness experiments.

**3.6 Occasional non-positive surface mass balance experiments**

So far, the surface mass balance values adopted in our experiments have been positive (corresponding to the accumulation zone). This limitation is sufficient for the usual topics relating to deep ice-core experiments where interpretation of ice-core data may become too complex. However, in order to provide a complete demonstration of the performance levels of numerical

385 age computations for more general cases, it is worthwhile to examine other cases. Although it may be considered pointless to examine steady negative mass balance cases because they simply mirror the steady positive cases presented above, the surface mass balance level adopted in this section is examined with zero or negative $a_\mathrm{L}$ in Eq. (57) and Fig. 5. One encountered difficulty is computing vertical velocity when the surface mass balance is negative. Strictly speaking, it is possible to apply negative $M_\mathrm{s}$ to Eqs. (52) and (53), but the validity of such a formulation may be questionable because it is based on an idealized

390 steady-state ice-sheet solution under positive surface mass balance conditions (e.g., Rybak and Huybrechts, 2003). However, for the sake of simplicity, the vertical velocity profiles in the current study are prescribed using the same set of equations for both positive and negative surface mass balances. This is considered to be sufficient, particularly for evaluations of scheme numerical performance levels.

Figure 14 is the result of transient experiments conducted under the square-wave surface mass balance of $a_\mathrm{L} = 0\,\mathrm{cm\,yr^{-1}}$,

395 while the other configuration is the same as Fig. 6 or 7. Figure 15 is a configuration with $a_\mathrm{L} = -1.5\,\mathrm{cm\,yr^{-1}}$. The thickness is fixed as 3000 m, the mass balances are $a_\mathrm{H} = 3.0\,\mathrm{cm\,yr^{-1}}$, $M_\mathrm{b} = 0$, and the phases are $P_\mathrm{H}, P_\mathrm{L} = 50, 50\,\mathrm{kyr}$.

[revised manuscript text omitted]

---

## Author Comment (AC1) · 19 Jun 2020

**Response to Reviewer comments**

We thank to the reviewers who provided precise and valuable feedbacks on our manuscript. We addressed all the points in the responses as follows. The reviewer comments are quoted in italic with some minor adjustments, and our responses to them follows them. We will be happy to submit a revised manuscript that reflects these changes, which significantly improves the quality of our manuscript.

**Response to Anonymous Reviewer ♯1**

We thank the reviewer for presenting several key points that will indeed improve the manuscript. We have addressed these concerns below.

**Summary**

*F. Saito et al. paper addresses the problem of numerical computation of ice age in ice sheet models. Indeed, calculation of ice age is the major challenge of ice sheet modeling in various applications beginning with the preliminary choice of potential target place for deep drilling of ice sheets and ending with the accurate interpretation of ice cores. The study area in Saito et al. manuscript is limited by a summit position of an ice sheet where the benchmark — an analytical solution for the ice age can be set. The authors examine two semi-Lagrangian RCIP schemes performance and compare results with more traditional Eulerian upwinding schemes for solving an advection equation.*

A perfect summary. Thanks a lot. We agree that the topic of the manuscript is limited to 1-d computation under summits, and believe that this is a necessary step for future extension of the new scheme to 3-d field computation, as the referee remarks in the following.

**General remarks**

*It should be noted that a family of RCIP schemes have been applied earlier in various problems of hydrodynamics, hydraulics etc., but their application for ice age calculation in ice sheet modeling is a novel and, perhaps, a promising approach. In reality, of course, we face with 3D problems of ice age computation, either when it is necessary to build the ice age field of the whole ice sheet or to construct a model chronology of a virtual ice core. From this point of view, the submitted research of Saito et al. may be considered as just an academic exercise comparing various numerical methods for highly idealized environmental conditions, which never occur in reality. Nevertheless, such kind of research are useful because they indicate possible pitfalls of rather traditional methods and introduce new approaches for solving tantalizing tasks in ice sheet modeling. In the 'Discussion and conclusion' section the authors reasonably point that the advection problem can be attributed not only to ice age calculation but also to calculation, for instance, of ice temperature. Anyway, further application of the RCIP method and its comparison with the Eulerian schemes in 3D will inevitably face with the choice of a true benchmark,*

*which are, indeed, absent except in case of visual calculation of annual ice layers in ice cores. Moreover, for model interpretation of ice cores a very effective back-tracing method was suggested (Huybrechts et al., Climate of the Past, 2007) which is a powerful tool for dating of ice cores using ice sheet modeling technique. In the 'Discussion and conclusion' section authors mention that their aim is to proceed with examination of the RCIP scheme in 3D. In this view, I think it would be reasonable to outline possible restrictions, challenges and limitations of future research.*

First of all, thank you for the evaluation and your precise understanding of this paper. Yes, this paper may be regarded as an exercise, however, as the reviewer kindly points out, we believe that this is a useful approach which we should not avoid when introducing a new scheme.

We agree in particular to the last remark. Citing Huybrechtes et al. (2007), we will introduce possible restrictions and limitations of this approach. The small but main advantage of RCIP method than the powerful back-tracing method is that it is a forward scheme — it is not necessary to record all the past velocity field during the simulation. RCIP may do a good job for preserving the flux information at the deposition (annual layer thickness in the case of dating), however, detection of 'points of origin' requires another technique, e.g., the back-tracing method itself. Thus we consider that the combination of the high precision forward scheme and the powerful backward scheme will be a good choice for ice-core dating issue. Thanks a lot for this comment.

*Another problem, which was not elucidated in the manuscript is the computational cost of application of different numerical schemes. I think it would be easy to do since all experiments were performed on the same computer facility. There is only short note on that (Line 239). The trade off between time of computation and accuracy in some cases may play for the simpler but faster method. In general, the manuscript is well structured, the figures are informative (except the note below concerning an a possible additional figure).*

Thanks a lot for pointing it out. Typically we computed $4 \times 7$ different configuration of 1d column in one run on an Intel Xeon E5-2609 6 core PC. The mean computational costs for one job in the case of 129 levels with the first-order upstream, the second-order, RCIP, RCIP with correction are 30, 28, 32, and 34 seconds, respectively. Those in the case of 513 levels are 338, 296, 364 and 392 seconds, respectively. These will be described in the text.

**Line by line comments**

*The title of the paper. The core of the paper is a set of comparisons between performance of the semi-Lagrangian RCIP schemes and the Eulerian once. Actually, there is nothing in the manuscript about ice sheet models. Therefore, it would be reasonable to be more precise in formulation of the title.*

Thanks a lot for your suggestion. Since the word 'ice-sheet' should be kept because the focus is on it, the last word is not necessary to satisfy your remark — 'Implementation of RCIP scheme and its performance for 1D age computations in ice-sheet'.

*Line 15. "$\cdots$ more generally in tracer transport $\cdots$". This statement is somewhat confusing. Dating of ice cores is not limited to tracer transport. This definition (tracer transport) may be attributed to Lagrangian or semi-Lagrangian methods only.*

All right, it is confusing. The block 'tracer transport...' will be deleted. Also the paragraph will be slightly adjusted according to this change.

***Section 1.1 and 1.2 The section lacks short general description of the semi-Lagrangian method in the context of its comparison with the pure Lagrangian and the Eulerian. Since the problem of interpolation is the most important in semi-Lagrangian schemes, it will be very much handful to make a (sketch) figure illustrating application of a 1-D semi-Lagrangian approach using definitions of the variables mentioned in the manuscript (arrival and departure points etc.). It would be also appropriate to address the reader to a classical paper (Stanoforth and Côté, 1991, Semi-Lagrangian integration schemes for atmospheric models: a review. Mon.Weather Rev., 119(9), 2206-2223.)***

This is a good point. A short general description will be inserted with a schematic figure to explain the design of semi-Lagrangian, arrival/departure points. The classical paper the reviewer mentioned will be cited.

***Line 61. Please comment on the first use of $g(x_j)$. What is it, what is the purpose of its introduction etc.***

The term $g(x_j)$ is abbreviation of function $g(x)$ (i.e. the spatial derivative of $f(x)$) at the grid-points $x_j$. A short description will be inserted.

***Line 187. To be precise, Rybak and Huybrechts (2003) did not employ semi-Lagrangian approach, but pure Lagrangian particle tracing.***

Thanks. 'Lagrangian' will be also inserted here.

***Line 394. "Figure 14 is the result$\cdots$" should be reformulated like, for instance, "Results of transient experiments are presented in Figure 14 $\cdots$". Same is in Line 395: "same as IN Fig. 6 AND 7." Same is in the next sentence.***

They will be reformulated as your suggestions.

***Line 459. Please, check equation for $\zeta$. What is $Z^{14}$? Please, explain why did you use this particular formula for the smooth discretization? What did you mean under "some trial and error". In my view, you should be more exact.***

Thanks a lot for point it out. The term $Z^{14}$ is $Z$ to the power of 14. We try the $\zeta$ formulation as $(Z + xZ^y)/(1 + x)$ with two parameters $x$ and $y$. The constrain we force was (i) $\zeta(Z = 0) = 0$ (ii) $\zeta(Z = 1) = 1$, (iii) $d\zeta/dZ > 0$. The formulation above is simple and satisfy these requirements. With varying $x$ and $y$, we found the formulation in the text is one of them to resolve the target annual layer thickness at the target depth. These will be described around here.

***Line 482. Please, indicate that your computations can be related to the summit points of ice sheets only, which are accepted stable throughout the time spell of numerical experiments.***

All right. The restriction of this study as you mention will be inserted.

**_Line 482. In my view, the fragment of the text "$\cdots$ ice-sheets under various configurations" is somewhat confusing. The results of the study are attributed to summits of ice sheets only, and their configurations have no any connection with the research._**

All right. The word 'various' is too much and will be removed. Together with the previous remark, the sentence will be more precise. Thanks a lot.

**Response to Reviewer ♯2**

We thank Shawn Marshall for a number of detailed review that significantly helped us to improve the quality of our manuscript. We have addressed these concerns below.

*The authors present a detailed examination of a novel (in ice sheets) interpolation scheme with promise for improved tracing of ice age as well as annual layer thickness reconstructions in ice sheets. This study focuses on 1D examples with scenarios (e.g. mass balance accumulation rates/vertical velocities) typical of the East Antarctic plateau, with direct relevance to ice core dating and age modelling.*

A perfect summary. Thanks a lot.

*The study is comprehensive, with superb attention to detail and to explaining the method and the mathematical implementation, such that this should provide a strong foundation for building on and for others that choose to adopt these methods. It is a valuable study, as age modelling or other passive tracer advection studies (e.g., isotopes, dust layers, or other chemical horizons) have not been given much attention in ice sheet studies in recent years, and are likely due for a resurgence as radar reconstructions are giving increasing detail on 3D ice sheet structure (e.g. McGregor et al., 2015); 3D tracer modelling offers an important avenue for improving and constraining ice sheet models. The methods introduced here should be seriously considered as an alternative to more 'classical' semi-Lagrangian interpolation schemes such as upwind differencing.*
*MacGregor, J. A., M. A. Fahnestock, G. A. Catania, J. D. Paden, S. Prasad Gogineni, S. K. Young, S. C. Rybarski, A. N. Mabrey, B. M. Wagman, and M. Morlighem (2015a), Radiostratigraphy and age structure of the Greenland Ice Sheet, J. Geophys. Res. Earth Surf., 120, 212–241, doi:10.1002/2014JF003215.*

Thanks a lot for such a positive evaluation.

*I am attaching a copy of the manuscript with several minor points. The English needs a bit of a double check throughout, for articles, but it is extremely well written and thorough, overall. I will confess that I did not work through the mathematical derivations carefully and have no experience with the RCIP or CIP techniques, so I cannot comment specifically on the rigour and appropriateness of this aspect of the manuscript, or on the novelty of the ideas (vs. e.g., existing implementations in other contexts such as atmospheric models). It is new and relevant to ice sheet modelling.*

Responses to all the minor points are appended at the next section. We are grateful to the reviewer for the careful and detail review. We believe that the manuscript will be significantly improved after modification following the suggestions and comments.

*There is a large number of figures, and it could be worthwhile to consider condensing the presentation of results a little. For instance, with new experiments/sensitivity tests after Figure 7, it could be possible to show only one result (of Figures 8 and 9, and of Figures 14 and 15; maybe elsewhere), while still discussing both experiments*

*in the text. I am also OK with the manuscript as is. Sometimes it is nice to see
everything laid out and presented, without relegating additional results to
supplements.*

All right, we agree. We will reduce the figures while keeping the text accordingly.

*A couple of suggestions for the authors' consideration:*
*The accumulation rates in the experiments are very low, typical of the East Antarctic
Plateau during the glacial period. I guess that it does not affect the performance of
the different interpolation/advection models, but am curious to confirm this for the
case of e.g. accumulation rates 10 times higher, more typical of Greenland. Also,
combined with this, high-amplitude, millennial-scale climate oscillations that are
typical of Greenland (D-O cycles). Are there specific recommendations or differences
in RCIP behaviour specific to these conditions?*

This is a good point.
Actually, as far as the shapes of normalized vertical velocity profile are identical, the normalized
shapes of the solutions are also identical. In other words, for example, the age solution under the
configuration of 30cm/yr surface mass balance, 0 basal mass balance, and 3000m ice thickness, has
the same normalized shape with the solution with that under 3cm/yr, 0 and 3000m, respectively.
Another example: Fig. 17 in the manuscript shows the results of annual layer thickness at **1000kyr**
in terms of **mm**, under the square wave surface mass balance between **3cm/yr** and **1.5cm/yr** with
total duration of **10,20**, and **50kyr**. These results can be, as they are, interpreted as annual layer
thickness at **100kyr** in terms of **10mm**, under those of **30cm/yr–15cm/yr** with the duration
**1**, **2**, and **5kyr**, respectively (i.e., corresponding 1/10 unit time.). Therefore roughly speaking,
the situation the referee is interested (millennial scale and typical Greenland) is already covered
by the same experiment. We have examined a part of sensitivity studies with 10 times higher
accumulation to confirm the above idea. The idea of scaling can be additional demonstration
worthwhile to present. We will insert the above discussion. Thanks a lot for this point.

*The model is developed specific to 1D age modelling in ice core settings (i.e. purely
vertical flow, positive surface mass balance). Extension to 3D is discussed near the
end, but would require consideration of positive (emergence) velocities, 3D flow fields,
and (typically) much lower horizontal gradients of ice age. This first comes up on
p.8, l.197, where the authors develop a formulation that assumes negative vertical
velocity throughout, which will not be compatible with 3D modelling. I appreciate that
the extension to 3D is for future study and we already have much to chew on with the
current presentation of ideas and results, but this discussion could be extended a bit
and I am curious about the author's opinion of whether the more complex RCIP type
of approach is warranted for the lower horizontal gradients in 3D interpolation
models.*

Yes, the negative mass balance experiment is just a demonstration and may not be compatible
with 3d situation. RCIP is in a sense merely a variation of semi-Lagrangian scheme: instead of
spatially increasing the number of grid-points for achieve higher-order interpolation, it does add a
field variable to solve (the gradient term). Therefore RCIP is essentially the same method with the

other higher-order semi-Lagrangian scheme, so we believe that this approach has a comparable characteristics with other semi-Lagrangian schemes that the many past studies have already presented and discussed.

In addition, the spatial gradient of age is not a diagnostic (passive) field but prognostic under the RCIP scheme. So we speculate that the precision of the spatial gradient is no worse (hopefully better) than the other higher-order semi-Lagrangian methods. However, RCIP performance on such lower gradients fields is itself one of big challenging topic, since many past RCIP papers concern mainly prevention of oscillation at high gradient fronts, as far as the authors study. Such an extension of this discussion will be inserted in the text. Thanks a lot for such a stimulating comment to improve our manuscript.

*Related to 3D models: the authors explore what would be considered as high vertical resolution in ice sheet models, from 129 to 513 vertical layers. This is much higher than many operational 3D ice sheet models that look at 3d (Stokes) solutions to the velocity field or Ice Age timescales: nz = 40 may be more typical. In the section on vertical resolution, it would be helpful to include an experiment with e.g. nz=33 to evaluation model performance at lower resolution. Does it further degrade the interpolation schemes and exaggerate the differences in modelled ice age, or do models converge as resolution declines?*

Actually, we have already performed, some of e.g., nz=33 cases. We have not studied the results in detail, but at least using lower resolution, the preservation of annual layer thickness is reduced at shallower depth. We will check the results more in detail and will discuss in the text.

*I am interested in the relatively strong results of the first-order upwind scheme. The authors do discuss this, but why is this consistently better than 2nd-order upwind schemes in almost all of the model experiments? In some cases it is of comparable performance to RCIP. Would the authors recommend always using 1st-order over 2nd-order upstream advection/interpolation models, and under what conditions might 1st-order advection schemes be adequate, vs. the RCIP-corr approach? A short discussion of 'practical suggestions' for eventual application of this technique in ice sheet models would be valuable.*

Yes, we were surprised to see that, too. The relatively better performance of the first-order upwind scheme is already presented in past studies (Greve et al 2002 cited at L282), which attributes to cancellation of errors between discretization and numerical diffusion. Moreover, as discussed in the manuscript, the design of mid-point rule on the first-order upwind scheme is not a *true* first-order scheme. Figure 3 presents that the solution by true first-order scheme (UP-1n) is worse by magnitude one than the second-order (UP-2), as we expected. It is possible to implement similar mid-point rule on the second-order scheme, which may improve the result of second-order Or, a different design of second-order scheme as Greve et al (2002). These may change the relative performances. Despite several difference of the past study, the result show similar performances qualitatively: the first-order results may better than the second-order except for the bottom.

Figure 3 also RCIP with upstream correction significantly improves the solution than RCIP without correction, which suggests an importance of non-constant velocity between the arrival and departure points to take into account. A mid-point rule formulation on the first-order scheme, in principle, corresponds to the former, with upstream correction.

The shape of normalized vertical velocity profile also may play a role for the relative performance. The bottom part is less *linear* than the upper part, thus the first-order approximation becomes worse. Some or all of these points lead the better performance of the first-order. We will extend these discussion in the text.

About practical suggestions. We considered that, as far as the annually layer thickness is not our concern, the classical upwind schemes are not a bad choice for dating. Using a first-order upwind scheme, a detail structure of surface mass balance history disappears very rapidly, but average features are quite well computed except for near the bottom. The second-order scheme preserves the history than the first, but without an effective slope limiter strange oscillation can strike the result as we demonstrated in the paper. We did not try any of such slope filters presented in the past studies because it is not our purpose, that is one of the reasons that second-order seems to be worse than the first. However, as far as the annually layer thickness is not a focus, the results by the second-order schemes are slightly better than those of the first-order throughout the experiment except for the most simple case (honestly, not better but more close to RCIP solution). Slope filters for higher-order upwind schemes on a non-uniform discretization is possible (as mentioned in the text citing Murman et al 2005), but rather complex than uniform discretization case. The conclusion of Greve et al (2002) already present such 'practical suggestions': the second-order, the TVDLF scheme with minmod filter, and even the first-order schemes are their proposal for dating. Our suggestion after this statement: if you expect good performance in annual layer thickness computation close to the bottom, using non-uniform discretization, then we strongly recommends to apply RCIP. We will cite their statement and our new suggestion will be inserted accordingly. Thanks a lot for pointing it out.

***Many thanks for this interesting contribution - I look forward to seeing the final version advance to GMD and push the research community forward.***

Again, thanks a lot for all of the fruitful comments which will definitely improve out manuscripts. We will try our best to brush up to meet your expectation.

**Minor points**

*page=1 areas. Or "the potential ... area."*

All right. Replaced with 'areas'.

*page=2 I feel compelled to note that this work on semi-Lagrangian tracer schemes was initiated in Clarke and Marshall (2002), and Tarasov and Peltier (2003) built off of this. Clarke et al. (2005) and Lhomme et al. (2005) built further, through the introduction of mass-balance based interpolation schemes to better address the age-depth relationship (as noted here) in several different Greenland cores. Clarke, G.K.C., Marshall, S.J., 2002. Isotopic balance of the Greenland Ice Sheet: modelled concentrations of water isotopes from 30,000 BP to present. Quaternary Science Reviews 21, 419–430*

Excellent. Thank you very much for the information. We will introduce Clark and Marshall (2005) here.

*page=2 performing a time-splitting....*

All right, will be inserted 'a', accordingly.

*page=2 on the time-splitting...*

All right, will be inserted 'the', accordingly.

*page=5 here, does $x$ refer to $x_{dep}$, per the line above? Or it would be more logical to me that $x_j$ in Eq (24) is $x_{dep}$, the fixed point of departure.*

The function of Eq. (24) is a linear formulation of $u$ on any $x$ between $x_j$ and $x_{dep}$. So, it is of coarse also $x = x_{dep}$ satisfies this formulation. In order to integrate along $x$ between $x_j$ and $x_{dep}$, we need such a formulation for $x$ in this range. This point will be more clearly described.

*page=6 delete "of"*

All right, will be deleted 'of' accordingly.

*page=7 Do you mean Eq. (32) here?*

Correct. Thanks a lot.

*page=7 is adopted*

Sure, not 'are' but 'is'.

*page=8 Clarke and Marshall, 2002*

Proper. Thanks a lot. Together with citation of the paper at page 2, this is introduced.

*page=8 although note that this will fail to accommodate full ice sheet conditions, e.g. in tracing layers into ablation zones*

You are right. This sentence somewhat overstates ice sheet dating computation. We will modify the statement according to your comment.

*page=10 computations*

All right, 'computations' accordingly.

*page=10 what do you mean, for p? for p=1?*

Sorry, the correct sentence is '... and setting $Ms = -Mb$ for arbitrarily $p$.' The word 'arbitrarily' was placed at wrong position.

*page=10 suggest $\sim$ rather than "around", here and later in this sentence*

All right. 'Around' hear and later will be replaced with '$\sim$'.

*page=12 We use a...*

All right. Rewrite as 'We use a uniform grid spacing of ...' accordingly.

*page=12 I am unclear on the units here - this is the error in years, perhaps, rather than kyr? At face values, it appears to have negative and positive biases of more than 10 kyr, but that is not consistent with (a)*

The unit is correct. Indeed the small oscillation at the bottom in (b) is obscured by the benchmark line in (a). Maybe zooming up will help the interpretation, which will be inserted.

*page=13 as a vertical*

All right. will be inserted 'a', accordingly.

*page=13 a very simple*

All right. will be inserted 'a', accordingly.

*page=14 These values are all fine but are extremely low for a lot of glaciological situations, e.g. in Greenland or WAIS divide, etc. Perhaps reflective of the glacial Antarctica plateau (3 cm/yr), but sensitivity tests could explore values and order of magnitude higher than this to be more representative of other ice sheet conditions.*

We definitely agree to this point. Please check our response in the previous section. Roughly speaking, the proper scaling of the result can be examined here. Also, the additional experiments with 10-times larger accumulation will be discussed here, which will draw more attention from readers. Thanks a lot for the suggestion.

*page=15 Did you explore sensitivity to nz? It might be good to discuss - nz=129 is greater resolution than many ice sheet modelling studies.*

Actually, yes. Using lower resolution, the preservation of annual layer thickness is reduced at shallower depth. This will be discussed in the text.

*page=20 of numerical performance of different schemes*

Thanks a lot. Corrected accordingly.

*page=23 (a) to (c) are backwards here, I think*

Yes, that's right. The labels above the figures are correct. Will be corrected.

*page=24 This is a great plot, but is hard to compare with the reference resolution in Figure 17 - perhaps each could be shown on the same y axis from e.g. 1000 to 2600 m?*

Great idea. We will extend the y-axis accordingly. Thanks a lot.

*page=25 a non-smooth grid*

All right. will be inserted 'a', accordingly.

---

## Author Response (AR1)

**Revision report and Author's response to the Reviewer comments**

We thank to the reviewers who provided precise and valuable feedbacks on our manuscript. We addressed all the points in the responses as follows. We are happy to submit the revised manuscript that reflects these changes, which significantly improves the quality of our manuscript.

The reviewer comments are quoted in italic with some minor editorial adjustments, and our responses to them follows them. All the comments are numbered (A0, B002, etc), and corresponding changes in the text and figures are annotated on the margins in the revised manuscript. Some modification are separated into multiple blocks and each of them is annotated with the same tag. Those changes corresponding to Reviewer ♯1 and to ♯2 are marked with red and blue, respectively, which include deletion/addition/replacement of the figures. Since numbers of figures and equations are changed, there are many small changes in these reference numbers, which are annotated with brown tags in the revised manuscript. Also, after including all the changes, English check is done by a company, which is also annotated with brown.

**Response to Anonymous Reviewer ♯1**

We thank the reviewer for presenting several key points that will indeed improve the manuscript. We have addressed these concerns below.

**Summary**

(A0)  *F. Saito et al. paper addresses the problem of numerical computation of ice age in ice sheet models. Indeed, calculation of ice age is the major challenge of ice sheet modeling in various applications beginning with the preliminary choice of potential target place for deep drilling of ice sheets and ending with the accurate interpretation of ice cores. The study area in Saito et al. manuscript is limited by a summit position of an ice sheet where the benchmark — an analytical solution for the ice age can be set. The authors examine two semi-Lagrangian RCIP schemes performance and compare results with more traditional Eulerian upwinding schemes for solving an advection equation.*

Thank you very much for your summary. We agree that the topic of the manuscript is limited to 1-d computation under summits, and believe that this is a necessary step for future extension of the new scheme to 3-d field computation, as the referee remarks in the following.

**General remarks**

(A1)  *It should be noted that a family of RCIP schemes have been applied earlier in various problems of hydrodynamics, hydraulics etc., but their application for ice age calculation in ice sheet modeling is a novel and, perhaps, a promising approach. In reality, of course, we face with 3D problems of ice age computation, either when it is necessary to build the ice age field of the whole ice sheet or to construct a model chronology of a virtual ice core. From this point of view, the submitted research of*

*Saito et al. may be considered as just an academic exercise comparing various numerical methods for highly idealized environmental conditions, which never occur in reality. Nevertheless, such kind of research are useful because they indicate possible pitfalls of rather traditional methods and introduce new approaches for solving tantalizing tasks in ice sheet modeling. In the 'Discussion and conclusion' section the authors reasonably point that the advection problem can be attributed not only to ice age calculation but also to calculation, for instance, of ice temperature. Anyway, further application of the RCIP method and its comparison with the Eulerian schemes in 3D will inevitably face with the choice of a true benchmark, which are, indeed, absent except in case of visual calculation of annual ice layers in ice cores. Moreover, for model interpretation of ice cores a very effective back-tracing method was suggested (Huybrechts et al., Climate of the Past, 2007) which is a powerful tool for dating of ice cores using ice sheet modeling technique. In the 'Discussion and conclusion' section authors mention that their aim is to proceed with examination of the RCIP scheme in 3D. In this view, I think it would be reasonable to outline possible restrictions, challenges and limitations of future research.*

First of all, thank you for the evaluation and your precise understanding of this paper. Yes, this paper may be regarded as an exercise, however, as the reviewer kindly points out, we believe that this is a useful approach which we should not avoid when introducing a new scheme.

We agree in particular to the last remark. Citing Huybrechtes et al. (2007), we introduced possible restrictions and limitations of this approach in the discussion section. The small but main advantage of RCIP method than the powerful back-tracing method is that it is a forward scheme — it is not necessary to record all the past velocity field during the simulation. RCIP may do a good job for preserving the flux information at the deposition (annual layer thickness in the case of dating), however, detection of 'points of origin' requires another technique, e.g., the back-tracing method itself. Thus we consider that the combination of the high precision forward scheme and the powerful backward scheme will be a good choice for ice-core dating issue. Thanks a lot for this comment.

(A2) *Another problem, which was not elucidated in the manuscript is the computational cost of application of different numerical schemes. I think it would be easy to do since all experiments were performed on the same computer facility. There is only short note on that (Line 239). The trade off between time of computation and accuracy in some cases may play for the simpler but faster method. In general, the manuscript is well structured, the figures are informative (except the note below concerning an a possible additional figure).*

Thanks a lot for pointing it out. Typically we computed $4 \times 7$ different configuration of 1d column in one run on an Intel Xeon E5-2609 6 core PC. The mean computational costs for one job in the case of 129 levels with the first-order upstream, the second-order, RCIP, RCIP with correction are 30, 28, 32, and 34 seconds, respectively. Those in the case of 513 levels are 338, 296, 364 and 392 seconds, respectively. These are described in the text in addition to the original text relating to the ratio of computing times.

**Line by line comments**

**(A000)** *The title of the paper. The core of the paper is a set of comparisons between performance of the semi-Lagrangian RCIP schemes and the Eulerian once. Actually, there is nothing in the manuscript about ice sheet models. Therefore, it would be reasonable to be more precise in formulation of the title.*

Thanks a lot for your suggestion. In particular for title, we agree to your suggestion. Since the word 'ice-sheet' should be kept because the focus is on it, only the last word is not necessary to satisfy your remark — 'Implementation of RCIP scheme and its performance for 1D age computations in ice-sheet'. Generally, however, 'an ice-sheet model' means various things: recently in most context it seems a dynamic ice-sheet *flow* model, but an ice-sheet *dating* model can be also an ice-sheet model. Also, as discussed shortly in the manuscript, we will extend RCIP implementation onto our ice-sheet *flow* model, so we keep most of the terminology in the main text.

**(A001)** *Line 15. "$\cdots$ more generally in tracer transport $\cdots$". This statement is somewhat confusing. Dating of ice cores is not limited to tracer transport. This definition (tracer transport) may be attributed to Lagrangian or semi-Lagrangian methods only.*

All right, we agree it was confusing. The block 'tracer transport...' are now deleted. Also the paragraph are slightly adjusted according to this change.

**(A002)** *Section 1.1 and 1.2 The section lacks short general description of the semi-Lagrangian method in the context of its comparison with the pure Lagrangian and the Eulerian. Since the problem of interpolation is the most important in semi-Lagrangian schemes, it will be very much handful to make a (sketch) figure illustrating application of a 1-D semi-Lagrangian approach using definitions of the variables mentioned in the manuscript (arrival and departure points etc.). It would be also appropriate to address the reader to a classical paper (Stanoforth and Côté, 1991, Semi-Lagrangian integration schemes for atmospheric models: a review. Mon.Weather Rev., 119(9), 2206-2223.)*

This is a good point. A short general descriptions are inserted in Sects 1.1 and 1.2 with a schematic figure to explain the design of semi-Lagrangian, arrival/departure points (new Figure 1). The classical paper the reviewer mentioned is cited also.

**(A003)** *Line 61. Please comment on the first use of $g(x_j)$. What is it, what is the purpose of its introduction etc.*

The term $g(x_j)$ is abbreviation of function $g(x)$ (i.e. the spatial derivative of $f(x)$) at the grid-points $x_j$. The definition of $g(x)$ was already documented before Eq.(4), but A short description is inserted again here in order to emphasize it.

**(A004)** *Line 187. To be precise, Rybak and Huybrechts (2003) did not employ semi-Lagrangian approach, but pure Lagrangian particle tracing.*

Thanks. 'Lagrangian' is also inserted here.

**(A005)** *Line 394. "Figure 14 is the result···" should be reformulated like, for instance, "Results of transient experiments are presented in Figure 14 ···". Same is in Line 395: "same as IN Fig. 6 AND 7." Same is in the next sentence.*

They are reformulated following your suggestions, with including the suggestion of the reviewer ii.

**(A006)** *Line 459. Please, check equation for $\zeta$. What is $Z^{14}$? Please, explain why did you use this particular formula for the smooth discretization? What did you mean under "some trial and error". In my view, you should be more exact.*

Thanks a lot for point it out. The term $Z^{14}$ is $Z$ to the power of 14. We try the $\zeta$ formulation as $(Z + \gamma Z^{\psi})/(1 + \gamma)$ with two parameters $\gamma$ and $\psi$. The constrain we force was (i) $\zeta(Z = 0) = 0$ (ii) $\zeta(Z = 1) = 1$, (iii) $d\zeta/dZ > 0$. The formulation above is simple and satisfy these requirements. With varying $x$ and $y$, we found the formulation in the text is one of them to resolve the target annual layer thickness at the target depth. These are now described around here.

**(A007)** *Line 482. Please, indicate that your computations can be related to the summit points of ice sheets only, which are accepted stable throughout the time spell of numerical experiments.*

All right. The restriction of this study as you mention is now inserted.

**(A008)** *Line 482. In my view, the fragment of the text "··· ice-sheets under various configurations" is somewhat confusing. The results of the study are attributed to summits of ice sheets only, and their configurations have no any connection with the research.*

All right. The word 'various' is too much and removed. Together with the previous remark, the sentence are written more precisely. Thanks a lot.

**Response to Reviewer ♯2**

We thank Shawn Marshall for a number of detailed review that significantly helped us to improve the quality of our manuscript. We have addressed these concerns below.

**(B0) *The authors present a detailed examination of a novel (in ice sheets) interpolation scheme with promise for improved tracing of ice age as well as annual layer thickness reconstructions in ice sheets. This study focuses on 1D examples with scenarios (e.g. mass balance accumulation rates/vertical velocities) typical of the East Antarctic plateau, with direct relevance to ice core dating and age modelling.***

Thank you very much for your summary.

**(B1) *The study is comprehensive, with superb attention to detail and to explaining the method and the mathematical implementation, such that this should provide a strong foundation for building on and for others that choose to adopt these methods. It is a valuable study, as age modelling or other passive tracer advection studies (e.g., isotopes, dust layers, or other chemical horizons) have not been given much attention in ice sheet studies in recent years, and are likely due for a resurgence as radar reconstructions are giving increasing detail on 3D ice sheet structure (e.g. McGregor et al., 2015); 3D tracer modelling offers an important avenue for improving and constraining ice sheet models. The methods introduced here should be seriously considered as an alternative to more 'classical' semi-Lagrangian interpolation schemes such as upwind differencing.***
***MacGregor, J. A., M. A. Fahnestock, G. A. Catania, J. D. Paden, S. Prasad Gogineni, S. K. Young, S. C. Rybarski, A. N. Mabrey, B. M. Wagman, and M. Morlighem (2015a), Radiostratigraphy and age structure of the Greenland Ice Sheet, J. Geophys. Res. Earth Surf., 120, 212–241, doi:10.1002/2014JF003215.***

Thanks a lot for such a positive evaluation.

**(B2) *I am attaching a copy of the manuscript with several minor points. The English needs a bit of a double check throughout, for articles, but it is extremely well written and thorough, overall. I will confess that I did not work through the mathematical derivations carefully and have no experience with the RCIP or CIP techniques, so I cannot comment specifically on the rigour and appropriateness of this aspect of the manuscript, or on the novelty of the ideas (vs. e.g., existing implementations in other contexts such as atmospheric models). It is new and relevant to ice sheet modelling.***

Responses to all the minor points are appended at the next section. We are grateful to the reviewer for the careful and detail review. We believe that the manuscript is now significantly improved after modification following the suggestions and comments.

**(B3) *There is a large number of figures, and it could be worthwhile to consider condensing the presentation of results a little. For instance, with new experiments/sensitivity tests after Figure 7, it could be possible to show only one result (of Figures 8 and 9, and of Figures 14 and 15; maybe elsewhere), while still***

*discussing both experiments in the text. I am also OK with the manuscript as is. Sometimes it is nice to see everything laid out and presented, without relegating additional results to supplements.*

All right, we agree. We reduce the figures while keeping the text accordingly. Figures 9, 13, 15 are now removed.

(B4) *A couple of suggestions for the authors' consideration:*
*The accumulation rates in the experiments are very low, typical of the East Antarctic Plateau during the glacial period. I guess that it does not affect the performance of the different interpolation/advection models, but am curious to confirm this for the case of e.g. accumulation rates 10 times higher, more typical of Greenland. Also, combined with this, high-amplitude, millennial-scale climate oscillations that are typical of Greenland (D-O cycles). Are there specific recommendations or differences in RCIP behaviour specific to these conditions?*

This is a good point.
Actually, as far as the shapes of normalized vertical velocity profile are identical, the normalized shapes of the solutions are also identical. In other words, for example, the age solution under the configuration of 30cm/yr surface mass balance, 0 basal mass balance, and 3000m ice thickness, has the same normalized shape with the solution with that under 3cm/yr, 0 and 3000m, respectively. Another example: Fig. 17 in the manuscript shows the results of annual layer thickness at **1000kyr** in terms of **mm**, under the square wave surface mass balance between **3cm/yr** and **1.5cm/yr** with total duration of **10,20**, and **50kyr**. These results can be, as they are, interpreted as annual layer thickness at **100kyr** in terms of **10mm**, under those of **30cm/yr–15cm/yr** with the duration **1, 2**, and **5kyr**, respectively (i.e., corresponding 1/10 unit time.). Therefore roughly speaking, the situation the referee is interested (millennial scale and typical Greenland) is already covered by the same experiment. We have examined a part of sensitivity studies with 10 times higher accumulation to confirm the above idea (figure is not shown). The idea of scaling can be additional demonstration worthwhile to present. We insert this idea in the introduction and result section, and also insert actual discussion in the last section. Thanks a lot for this point.

(B5) *The model is developed specific to 1D age modelling in ice core settings (i.e. purely vertical flow, positive surface mass balance). Extension to 3D is discussed near the end, but would require consideration of positive (emergence) velocities, 3D flow fields, and (typically) much lower horizontal gradients of ice age. This first comes up on p.8, l.197, where the authors develop a formulation that assumes negative vertical velocity throughout, which will not be compatible with 3D modelling. I appreciate that the extension to 3D is for future study and we already have much to chew on with the current presentation of ideas and results, but this discussion could be extended a bit and I am curious about the author's opinion of whether the more complex RCIP type of approach is warranted for the lower horizontal gradients in 3D interpolation models.*

Yes, the negative mass balance experiment is just a demonstration and may not be compatible with 3d situation. RCIP is in a sense merely a variation of semi-Lagrangian scheme: instead of

spatially increasing the number of grid-points for achieve higher-order interpolation, it does add a field variable to solve (the gradient term). Therefore RCIP is essentially the same method with the other higher-order semi-Lagrangian scheme, so we believe that this approach has a comparable characteristics with other semi-Lagrangian schemes that the many past studies have already presented and discussed. In addition, the spatial gradient of age is not a diagnostic (passive) field but prognostic under the RCIP scheme. So we speculate that the precision of the spatial gradient is no worse (hopefully better) than the other higher-order semi-Lagrangian methods. Such an extension of this discussion are inserted in the text. Thanks a lot for such a stimulating comment to improve our manuscript.

(B6) *Related to 3D models: the authors explore what would be considered as high vertical resolution in ice sheet models, from 129 to 513 vertical layers. This is much higher than many operational 3D ice sheet models that look at 3d (Stokes) solutions to the velocity field or Ice Age timescales: nz = 40 may be more typical. In the section on vertical resolution, it would be helpful to include an experiment with e.g. nz=33 to evaluation model performance at lower resolution. Does it further degrade the interpolation schemes and exaggerate the differences in modelled ice age, or do models converge as resolution declines?*

Actually, we have already performed, some of e.g., nz=33 cases. An example result is inserted in the manuscript (new Figure 17). Also, Fig. 14 (which was Fig.16 in the first manuscript) now contains a lines corresponding to nz=33. The preservation of annual layer thickness is reduced at shallower depth.

(B7) *I am interested in the relatively strong results of the first-order upwind scheme. The authors do discuss this, but why is this consistently better than 2nd-order upwind schemes in almost all of the model experiments? In some cases it is of comparable performance to RCIP. Would the authors recommend always using 1st-order over 2nd-order upstream advection/interpolation models, and under what conditions might 1st-order advection schemes be adequate, vs. the RCIP-corr approach? A short discussion of 'practical suggestions' for eventual application of this technique in ice sheet models would be valuable.*

Yes, we were surprised to see that, too. The relatively better performance of the first-order upwind scheme is already presented in past studies (Greve et al 2002 cited at L282), which attributes to cancellation of errors between discretization and numerical diffusion. Moreover, as discussed in the manuscript, the design of mid-point rule on the first-order upwind scheme is not a *true* first-order scheme. Figure 3 presents that the solution by true first-order scheme (UP-1n) is worse by magnitude one than the second-order (UP-2), as we expected. It is possible to implement similar mid-point rule on the second-order scheme, which may improve the result of second-order Or, a different design of second-order scheme as Greve et al (2002). These may change the relative performances. Despite several difference of the past study, the result show similar performances qualitatively: the first-order results may better than the second-order except for the bottom.

Figure 3 also RCIP with upstream correction significantly improves the solution than RCIP without correction, which suggests an importance of non-constant velocity between the arrival and departure points to take into account. A mid-point rule formulation on the first-order scheme, in principle, corresponds to the former, with upstream correction.

The shape of normalized vertical velocity profile also may play a role for the relative performance. The bottom part is less *linear* than the upper part, thus the first-order approximation becomes worse. Some or all of these points lead the better performance of the first-order. We extend these discussion in the manuscript.

About practical suggestions. We considered that, as far as the annually layer thickness is not our concern, the classical upwind schemes are not a bad choice for dating. Using a first-order upwind scheme, a detail structure of surface mass balance history disappears very rapidly, but average features are quite well computed except for near the bottom. The second-order scheme preserves the history than the first, but without an effective slope limiter strange oscillation can strike the result as we demonstrated in the paper. We did not try any of such slope filters presented in the past studies because it is not our purpose, that is one of the reasons that second-order seems to be worse than the first. However, as far as the annually layer thickness is not a focus, the results by the second-order schemes are slightly better than those of the first-order throughout the experiment except for the most simple case (honestly, not better but more close to RCIP solution). Slope filters for higher-order upwind schemes on a non-uniform discretization is possible (as mentioned in the text citing Murman et al 2005), but rather complex than uniform discretization case. The conclusion of Greve et al (2002) already present such 'practical suggestions': the second-order, the TVDLF scheme with minmod filter, and even the first-order schemes are their proposal for dating. Our suggestion after this statement: if you expect good performance in annual layer thickness computation close to the bottom, using non-uniform discretization, then we strongly recommends to apply RCIP. We cite their statement and our new suggestion are inserted accordingly. Thanks a lot for pointing it out.

(B8)  *Many thanks for this interesting contribution - I look forward to seeing the final version advance to GMD and push the research community forward.*

Again, thanks a lot for all of the fruitful comments which definitely improve out manuscripts.

**Minor points**

(B000)  *page=1 areas. Or "the potential ... area."*

All right. Replaced with 'areas'.

(B001)  *page=2 I feel compelled to note that this work on semi-Lagrangian tracer schemes was initiated in Clarke and Marshall (2002), and Tarasov and Peltier (2003) built off of this. Clarke et al. (2005) and Lhomme et al. (2005) built further, through the introduction of mass-balance based interpolation schemes to better address the age-depth relationship (as noted here) in several different Greenland cores. Clarke, G.K.C., Marshall, S.J., 2002. Isotopic balance of the Greenland Ice Sheet: modelled concentrations of water isotopes from 30,000 BP to present. Quaternary Science Reviews 21, 419–430*

Good point. Thank you very much for the information. We introduce Clark and Marshall (2005) here and other places.

(B002)  *page=2 performing a time-splitting....*

All right, inserted 'a', accordingly.

(B003) **page=2 on the time-splitting...**

All right, inserted 'the', accordingly.

(B004) **page=5 here, does $x$ refer to $x_{dep}$, per the line above? Or it would be more logical to me that $x_j$ in Eq (24) is $x_{dep}$, the fixed point of departure.**

After posting of the author's comment, we found an error on Eq. (24), which may confuse the reviewer. The function of Eq. (24) is a linear formulation of $u$ on any $x$ between $x_j$ and $x_{jj+1}$, not between $x_j$ and $x_{jj+1}$ as in the original submission. So, any $x$ between $x_j$ and $x_{jj+1}$, including $x = x_{dep}$ satisfies this formulation, as the reviewer pointed out. In order to integrate along $x$ between $x_j$ and $x_{dep}$, we need such a formulation for $x$ in this range. Correcting the error, we believe that this point is more clearly described now. We are sorry about the mistake, and thanks a lot to the reviewer for pointing it out.

(B005) **page=6 delete "of"**

All right, deleted 'of' accordingly.

(B006) **page=7 Do you mean Eq. (32) here?**

Correct. Thanks a lot.

(B007) **page=7 is adopted**

Thanks a lot. We modified the noun (representations) instead.

(B008) **page=8 Clarke and Marshall, 2002**

Thanks a lot. Together with citation of the paper at page 2, this is introduced.

(B009) **page=8 although note that this will fail to accommodate full ice sheet conditions, e.g. in tracing layers into ablation zones**

You are right. This sentence somewhat overstates ice sheet dating computation. We modify the statement according to your comment.

(B010) **page=10 computations**

All right, 'computations' accordingly.

(B011) **page=10 what do you mean, for $p$? for $p=1$?**

Sorry, the correct sentence is '... and setting $Ms = -Mb$ for arbitrarily $p$.' The word 'arbitrarily' was placed at wrong position.

(B012) **page=10 suggest $\sim$ rather than "around", here and later in this sentence**

All right. 'Around' hear and later are be replaced with '∼'.

**(B013)  *page=12 We use a...***

All right. Rewritten as 'We use a uniform grid spacing of ...' accordingly.

**(B014)  *page=12 I am unclear on the units here - this is the error in years, perhaps, rather than kyr? At face values, it appears to have negative and positive biases of more than 10 kyr, but that is not consistent with (a)***

The unit is correct. Indeed the small oscillation at the bottom in (b) is obscured by the benchmark line in (a). Zooming up of the figure (c) is inserted.

**(B015)  *page=13 as a vertical***

All right, inserted 'a', accordingly.

**(B016)  *page=13 a very simple***

All right, inserted 'a', accordingly.

**(B017)  *page=14 These values are all fine but are extremely low for a lot of glaciological situations, e.g. in Greenland or WAIS divide, etc. Perhaps reflective of the glacial Antarctica plateau (3 cm/yr), but sensitivity tests could explore values and order of magnitude higher than this to be more representative of other ice sheet conditions.***

We definitely agree to this point. Please check our response above (B4). Roughly speaking, the proper scaling of the result are examined. Also, the additional experiments with 10-times larger accumulation are discussed here, which may draw more attention from readers. Thanks a lot for the suggestion.

**(B018)  *page=15 Did you explore sensitivity to nz? It might be good to discuss - nz=129 is greater resolution than many ice sheet modelling studies.***

Actually, yes (see response to B6). Using lower resolution, the preservation of annual layer thickness is reduced at shallower depth. This is discussed in the text.

**(B019)  *page=20 of numerical performance of different schemes***

Thanks a lot. Corrected accordingly, with inserting a word 'levels'.

**(B020)  *page=23 (a) to (c) are backwards here, I think***

Yes, that's right. Corrected accordingly. The labels above the figures are correct.

**(B021)  *page=24 This is a great plot, but is hard to compare with the reference resolution in Figure 17 - perhaps each could be shown on the same y axis from e.g. 1000 to 2600 m?***

Great idea. We extend the y-axis accordingly. Thanks a lot.

**(B022)  *page=25 a non-smooth grid***

All right. inserted 'a', accordingly.

[revised manuscript text omitted]

---

## Author Response (AR2)

**Revision report and Author's response to the Topical Editor's comments**

We thank to the editor to provide good feedbacks on our manuscript. We addressed all the points in the responses as follows. We are happy to submit the revised manuscript that reflects these changes, which improved the quality of our manuscript and meet the GMD standards.

**(E0)** *Thank you for this interesting paper offering a very thorough analysis of an alternative numerical scheme for Eulerian advection problems in ice sheets. Based on the referee comments and my own evaluation the paper is almost ready for publication. Contentwise your revisions are fine. There are however a few technical issues you need to consider before final acceptance.*

Thank you for the evaluation of this paper. We are very happy to hear that the all our revisions in the previous manuscript are fine. Now we have revised the manuscript according to the two technical issues. Both issues are a little bit related each other, so here we summarize the revision first, followed by each comment.

About the public release of our software. From the conclusion, the copyright holders have agreed to make it public. Thanks a lot for your advice, which helped to make the decision. We have opened not only our implementation of the RCIP scheme, but also the full package of software code we used in this paper. The full package has been already opened under the Apache license version 2.0, on a github repository at `https://github.com/saitofuyuki/icies2`. The exact version is tagged as `archive_gmd-2020`. In addition, we already reserved the zenodo DOI (`https://doi.org/10.5281/zenodo.4034557`) to permanently archive the exact version of the full code and the scripts. The zenodo status will be changed to 'open access' when the paper is accepted.

Then, about the name and version number of the model. We have named the version of full package of 1d dating model as 'IcIES-2/JP version 0'. This is just a small subset of IcIES-2, which is 3D thermodynamic SIA/SSA ice sheet model, preparing to be published. We can add the name and version number in the title as other GMD development and technical articles, however, we are afraid that it will involve not a small change in the main text. It would look like a typical description paper of IcIES-2/JP, and we would need to describe IcIES-2/JP rather in detail in the main text. In this revision we do not add the model name and version number to the title, instead we explicitly mentioned it in the code availability section. We appreciate if you agree this minimum change, but of course we are happy to receive any other suggestions about this issue.

In summary, we revised the title and code availability section following the both comments. Moreover, there is one small editorial correction (which is marked in the text, L27).

**(E1)** *1. The title really sounds a bit awkward. If you do not want to add 'models' you should at least change 'ice-sheet' into 'ice sheets'. Usually the title of a development and technical paper contains a name for the code and its version number. Is this available for the RCIP code?*

Thanks a lot for your comment. We discussed this issue again among the co-authors, and we decided to revert the title as the original. Ice-core dating can be also a ice-sheet model, (as mentioned in our previous response), we believe it is natural to include a word 'models' in the title of this paper.

*(E2)  2. You submitted this paper as a development and technical paper. According to the journal guidelines for this paper type the code should normally be made public on a persistent public archive. Availibility on request is generally not accepted. I understand from the code availibility paragraph that the exact version of the code is protected under JAMSTEC copyright. Please try to make at least the core of the RCIP scheme fully publicly available.*
*For more background on the above technical issues, please read carefully the requirements for each paper type under the 'About' tab on the GMD website.*

Thanks a lot for the suggestion. We seriously discussed this issue, and the full code is now publicly available.

[revised manuscript text omitted]